# BALANCING TRAINING TIME VS. PERFORMANCE WITH BAYESIAN EARLY PRUNING

## ABSTRACT

Pruning is an approach to alleviate overparameterization of *deep neural network* (DNN) by zeroing out or pruning DNN elements with little to no efficacy at a given task. In contrast to related works that do pruning before or after training, this paper presents a novel method to perform *early* pruning of DNN elements (e.g., neurons or convolutional filters) *during the training process* while preserving performance upon convergence. To achieve this, we model the future efficacy of DNN elements in a Bayesian manner conditioned upon efficacy data collected during the training and prune DNN elements which are predicted to have low efficacy after training completion. Empirical evaluations show that the proposed Bayesian early pruning improves the computational efficiency of DNN training. Using our approach we are able to achieve a $48.6\%$ faster training time for ResNet-50 on ImageNet to achieve a validation accuracy of $72.5\%$.

## 1 INTRODUCTION

*Deep neural networks* (DNNs) are known to be overparameterized (Allen-Zhu et al., 2019) as they usually have more learnable parameters than needed for a given learning task. So, a trained DNN contains many ineffectual parameters that can be safely pruned or zeroed out with little/no effect on its predictive accuracy. *Pruning* (LeCun et al., 1989) is an approach to alleviating overparameterization of a DNN by identifying and removing its ineffectual parameters while preserving its predictive accuracy on the validation/test dataset. Pruning is typically applied to the DNN *after training* to speed up *testing-time evaluation*. For standard image classification tasks with MNIST, CIFAR-10, and ImageNet datasets, it can reduce the number of learnable parameters by up to $50\%$ or more while maintaining test accuracy (Han et al., 2015; Li et al., 2017; Molchanov et al., 2017).

In particular, the overparameterization of a DNN also leads to considerable training time being wasted on those DNN elements (e.g., connection weights, neurons, or convolutional filters) which are eventually ineffectual after training and can thus be safely pruned. Our work in this paper considers *early pruning* of such DNN elements by identifying and removing them *throughout the training process* instead of after training.[1] As a result, this can significantly reduce the time incurred by the training process without compromising the final test accuracy (upon convergence) much.

Recent work (Section 5) in foresight pruning (Lee et al., 2019; Wang et al., 2020) show that pruning heuristics applied at initialization work well to prune connection weights without significantly degrading performance. In contrast to these work, we prune throughout the training procedure, which improves performance after convergence of DNNs, albeit with somewhat longer training times.

In this work, we pose early pruning as a constrained optimization problem (Section 3.1). A key challenge in the optimization is accurately modeling the future efficacy of DNN elements. We achieve this through the use of *multi-output Gaussian process* which models the *belief* of future efficacy conditioned upon efficacy measurements collected during training (Section 3.2). Although the posed optimization problem is NP-hard, we derive an efficient *Bayesian early pruning* (BEP) approximation algorithm, which appropriately balances the inherent training time vs. performance tradeoff in pruning prior to convergence (Section 3.3). Our algorithm relies on a measure of network element efficacy, termed saliency (LeCun et al., 1989). The development of saliency functions is an active area of research with no clear optimal choice. To accomodate this, our algorithm is agnostic, and therefore

---

[1]In contrast, foresight pruning (Wang et al., 2020) removes DNN elements *prior* to the training process.

flexible, to changes in saliency function. We use BEP to prune neurons and convolutional filters to achieve practical speedup during training (Section 4).[2] Our approach also compares favorably to state-of-the-art works such as *SNIP* (Lee et al., 2019), *GraSP* (Wang et al., 2020), and *momentum based dynamic sparse reparameterization* (Dettmers & Zettlemoyer, 2019).

## 2 PRUNING

Consider a dataset of $D$ training examples $\mathcal{X} = \{\mathbf{x}_1, \ldots, \mathbf{x}_D\}, \mathcal{Y} = \{y_1, \ldots, y_D\}$ and a neural network $\mathcal{N}_{\boldsymbol{v}_t}$ parameterized by a vector of $M$ *pruneable* network elements (e.g. weight parameters, neurons, or convolutional filters) $\boldsymbol{v}_t \triangleq [v_t^a]_{a=1,\ldots,M}$, where $\boldsymbol{v}_t$ represent the network elements after $t$ iterations of stochastic gradient descent (SGD) for $t = 1, \ldots, T$. Let $\mathcal{L}(\mathcal{X}, \mathcal{Y}; \mathcal{N}_{\boldsymbol{v}_t})$ be the loss function for the neural network $\mathcal{N}_{\boldsymbol{v}_t}$. Pruning aims at refining the network elements $\boldsymbol{v}_t$ given some sparsity budget $B$ and preserving the accuracy of the neural network after convergence (i.e., $\mathcal{N}_{\boldsymbol{v}_T}$), which can be stated as a constrained optimization problem (Molchanov et al., 2017):

$$\min_{\boldsymbol{m} \in \{0,1\}^M} |\mathcal{L}(\mathcal{X}, \mathcal{Y}; \mathcal{N}_{\boldsymbol{m} \odot \boldsymbol{v}_T}) - \mathcal{L}(\mathcal{X}, \mathcal{Y}; \mathcal{N}_{\boldsymbol{v}_T})| \quad \text{s.t.} \quad ||\boldsymbol{m}||_0 \leq B \qquad (1)$$

where $\odot$ is the Hadamard product and $\boldsymbol{m}$ is a pruning mask. Note that we abuse the Hadamard product for notation simplicity: for $a = 1, .., M$, $m^a \times v_T^a$ corresponds to pruning $v_T^a$ if $m^a = 0$, and keeping $v_T^a$ otherwise. Pruning a network element refers to zeroing the network element or the weight parameters which compute the network element. Any weight parameters which reference the output of the pruned network element are also zeroed since the element outputs a constant 0.

The above optimization problem is difficult due to the NP-hardness of combinatorial optimization. This leads to the approach of using saliency function $s$ which measures efficacy of network elements at minimizing the loss function. A network element with small saliency can be pruned since it's not *salient* in minimizing the loss function. Consequently, pruning can be done by maximizing the saliency of the network elements given the sparsity budget $B$:

$$\max_{\boldsymbol{m} \in \{0,1\}^M} \sum_{a=1}^M m^a s(a; \mathcal{X}, \mathcal{Y}, \mathcal{N}_{\boldsymbol{v}_T}, \mathcal{L}) \quad \text{s.t.} \quad ||\boldsymbol{m}||_0 \leq B \qquad (2)$$

where $s(a; \mathcal{X}, \mathcal{Y}, \mathcal{N}_{\boldsymbol{v}_T}, \mathcal{L})$ measures the saliency of $v_T^a$ at minimizing $\mathcal{L}$ after convergence through $T$ iterations of SGD. The above optimization problem can be efficienctly solved by selecting the $B$ most *salient* network elements in $\boldsymbol{v}_T$.

The construction of the saliency function has been discussed in many existing works: Some approaches derived the saliency function from first-order (LeCun et al., 1989; Molchanov et al., 2017) and second-order (Hassibi & Stork, 1992; Wang et al., 2020) Taylor series approximations of $\mathcal{L}$. Other common saliency functions include $L_1$ (Li et al., 2017) or $L_2$ (Wen et al., 2016) norm of the network element weights, as well as mean activation (Polyak & Wolf, 2015). In this work, we use a first-order Taylor series approximation saliency function defined for neurons and convolutional filters[3] (Molchanov et al., 2017), however our approach remains flexible to arbitrary choice of saliency function on a plug-n-play basis.

## 3 BAYESIAN EARLY PRUNING

### 3.1 PROBLEM STATEMENT

As has been mentioned before, existing pruning works based on the saliency function are typically done after the training convergence (i.e., (2)) to speed up the testing-time evaluation, which waste considerable time on training these network elements which will eventually be pruned. To resolve this issue, We extend the pruning problem definition (2) along the temporal dimension, allowing network elements to be pruned during the training process consisting of $T$ iterations of SGD.

---

[2] Popular deep learning libaries do not accelerate sparse matrix operations over dense matrix operations. Thus, pruning network connections cannot be easily capitalized upon with performance improvements. It is also unclear whether moderately sparse matrix operations (i.e., operations on matrices generated by connection pruning) can be significantly accelerated on massively parallel architectures such as GPUs (see Yang et al. (2018) Fig. 7). See Section 5 in Buluç & Gilbert (2008) for challenges in parallel sparse matrix multiplication.

[3] Implementation details of this saliency function can be found in Appendix A.1.

Let $s_t^a \triangleq s(a; \mathcal{X}, \mathcal{Y}, \mathcal{N}_{\boldsymbol{v}_t}, \mathcal{L})$ be a random variable which denotes the saliency of network element $v_t^a$ after $t$ iterations of SGD, $\boldsymbol{s}_t \triangleq [s_t^a]_{a=1,\ldots,M}$ for $t = 1, \ldots, T$, and $\boldsymbol{s}_{\tau_1:\tau_2} \triangleq [\boldsymbol{s}_t]_{t=\tau_1,\ldots,\tau_2}$ be a vector of saliency of all the network elements between iterations $\tau_1$ and $\tau_2$. Our early pruning algorithm is designed with the goal of maximizing the saliency of the *unpruned* elements after iteration $T$, yet allowing for pruning at each iteration $t$ given some computational budget $B_{t,c}$ for $t = 1, \ldots, T$:

$$\rho_T(\boldsymbol{m}_{T-1}, B_{T,c}, B_s) \triangleq \max_{\boldsymbol{m}_T} \boldsymbol{m}_T \cdot \boldsymbol{s}_T \text{ (3a)}$$
$$\text{s.t.} \qquad ||\boldsymbol{m}_T||_0 \leq B_s \text{ (3b)} \quad \boldsymbol{m}_T \stackrel{.}{\leq} \boldsymbol{m}_{T-1} \text{ (3c)} \quad \mathrm{B}_{T,c} \geq 0 \text{ (3d)}$$

$$\rho_t(\boldsymbol{m}_{t-1}, B_{t,c}, B_s) \triangleq \max_{\boldsymbol{m}_t} \mathbb{E}_{p(\boldsymbol{s}_{t+1}|\tilde{\boldsymbol{s}}_{1:t})} \big[ \rho_{t+1}(\boldsymbol{m}_t, B_{t,c} - ||\boldsymbol{m}_t||_0, B_s) \big] \text{ (4a)}$$
$$\text{s.t.} \qquad \boldsymbol{m}_t \stackrel{.}{\leq} \boldsymbol{m}_{t-1} \text{ (4b)}$$

where $B_s$ is the trained network *sparsity* budget, $\tilde{\boldsymbol{s}}_{1:t}$ is a vector of observed values for $\boldsymbol{s}_{1:t}$, $\boldsymbol{m}_0$ is an $M$-dimensional 1's vector, and $\boldsymbol{m}_t \stackrel{.}{\leq} \boldsymbol{m}_{t-1}$ represents an element-wise comparison between $\boldsymbol{m}_t$ and $\boldsymbol{m}_{t-1}$: $m_t^a \leq m_{t-1}^a$ for $a = 1, \ldots, M$. At each iteration $t$, the saliency $\boldsymbol{s}_t$ is observed and $\boldsymbol{m}_t \in \{0, 1\}^M$ in $\rho_t$ represents a pruning decision performed to maximize the *expectectation* of $\rho_{t+1}$ conditioned upon saliency measurements $\boldsymbol{s}_{1:t}$ collected up to and including iteration $t$. This recursive structure terminates with base case $\rho_T$ where the saliency of the *unpruned* elements is maximized after $T$ iterations of training.

In the above early pruning formulation[4], constraints (3c) and (4b) ensure pruning is performed in a practical manner whereby once a network element is pruned, it can no longer be recovered in a later training iteration. We define a trained network sparsity budget, $B_s$ (3b), which may differ significantly from initial network size $||m_0||_0$ (e.g., in the case where the network is trained on GPUs, but deployed on resource constrained edge or mobile devices). We also constrain a total computational effort budget $B_{t,c}$ which is reduced per training iteration $t$ by the number of unpruned network elements $||\boldsymbol{m}_t||_0$. We constrain $B_{T,c} \geq 0$ (3d) to ensure training completion within the specified computational budget. Here we assume that a more sparse pruning mask $\boldsymbol{m}_t$ corresponds to lower computational effort during training iteration $t$ due to updating fewer network elements. Finally, (3a) maximizes the saliency with a pruning mask $\boldsymbol{m}_T$ constrained by a *sparsity* budget $B_s$ (3b). Our early pruning formulation balances the saliency of network elements after convergence against the total computational effort to train such network (i.e., $\boldsymbol{m}_T \cdot \boldsymbol{s}_T$ vs. $\sum_{t=1}^{T} ||\boldsymbol{m}_t||_0$). This appropriately captures the balancing act of training-time early pruning whereby the computational effort is saved by *early pruning* network elements while preserving the saliency of the remaining network elements after convergence.

### 3.2 MODELING THE SALIENCY WITH MULTI-OUTPUT GAUSSIAN PROCESS

To solve the above early pruning problem, we need to model the belief $p(\boldsymbol{s}_{1:T})$ of the saliency for computing the predictive belief $p(\boldsymbol{s}_{t+1:T}|\tilde{\boldsymbol{s}}_{1:t})$ of the future saliency in (4a). At the first glance, one may consider to decompose the belief: $p(\boldsymbol{s}_{1:T}) \triangleq \prod_{a=1}^{M} p(\boldsymbol{s}_{1:T}^a)$ and model the saliency $\boldsymbol{s}_{1:T}^a \triangleq [s_t^a]_{t=1,\ldots,T}$ of each network element independently. Such independent models, however, ignore the *co-adaptation* and *co-evolution* of the network elements which have been shown to be a common occurrence in DNN (Hinton et al., 2012; Srivastava et al., 2014; Wang et al., 2020). Also, modeling the correlations between the saliency of different network elements *explicitly* is non-trivial since considerable feature engineering is needed for representing diverse network elements such as neurons, connections, or convolutional filters.

To resolve such issues, we use *multi-output Gaussian process* (MOGP) to jointly model the belief $p(\boldsymbol{s}_{1:T})$ of all saliency measurements. To be specific, we assume that the saliency $s_t^a$ of the $a$-th network element at iteration $t$ is a linear mixture[5] of $Q$ independent latent functions $\{u_q(t)\}_{q=1}^{Q}$: $s_t^a \triangleq \sum_{q=1}^{Q} \gamma_q^a u_q(t)$. As shown in (Álvarez & Lawrence, 2011), if each $u_q(t)$ is an independent GP with *prior* zero mean and covariance $k_q(t, t')$, then the resulting distribution over $p(\boldsymbol{s}_{1:T})$ is a multivariate Gaussian distribution with prior zero mean and covariance determined by the mixing

---

[4]In contrast to PruneTrain (Lym et al., 2019), our problem definition balances training time vs. performance under an additional constraint on the trained network size (3b). We discuss this further in Section 5.

[5]Among the various types of MOGPs (see Álvarez & Lawrence (2011) for a detailed review.), we choose this linear model such that the correlations between $s_t^a$ and $s_{t'}^{a'}$ can be computed analytically.

weights: $\text{cov}[s_t^a, s_{t'}^{a'}] = \sum_{q=1}^{Q} \gamma_q^a \gamma_q^{a'} k_q(t, t')$. This explicit covariance between $s_t^a$ and $s_{t'}^{a'}$ helps to exploit the co-evolution and co-adaptation of network elements within the neural networks.

To capture the horizontal asymptote trend of $s_1^a, \ldots, s_T^a$ as visualized in Appendix A.2, we turn to a kernel used for modeling decaying exponential curves known as the "exponential kernel" (Swersky et al., 2014) and set $k_q(t, t') \triangleq \frac{\beta_q^{\alpha_q}}{(t+t'+\beta_q)^{\alpha_q}}$ where $\alpha_q$ and $\beta_q$ are hyperparameters of MOGP and can be learned via maximum likelihood estimation (Álvarez & Lawrence, 2011). Then, given a vector of observed saliency $\tilde{s}_{1:t}$, the MOGP regression model can provide a Gaussian predictive distribution for any future saliency $s_{t'}$. Thus, the predictive mean $\mu_{t'|1:t}^a \triangleq \mathbb{E}[s_{t'}^a \mid \tilde{s}_{1:t}]$ of the saliency $s_{t'}^a$ and the predictive (co)variance $\sigma_{t'|1:t}^{aa'} \triangleq \text{cov}[s_{t'}^a, s_{t'}^{a'} \mid \tilde{s}_{1:t}]$ between the saliency $s_{t'}^a$ and $s_{t'}^{a'}$ can be computed analytically, as detailed in Appendix A.3.

### 3.3 EARLY PRUNING ALGORITHM

Solving the above optimizing problem (3) and (4) is difficult due to the interplay between $[\boldsymbol{m}_{t'}]_{t'=t,\ldots,T}$, $[B_{t',c}]_{t'=t,\ldots,T}$, and $\boldsymbol{m}_T \cdot \boldsymbol{s}_T$. Instead, we consider a simplification of the above problem by only considering solutions of the form $\boldsymbol{m}_{T-1} = \boldsymbol{m}_{T-2} = \ldots = \boldsymbol{m}_t$ which yields[6]:

$$\hat{\rho}_t(\boldsymbol{m}_{t-1}, B_{t,c}, B_s) \triangleq \max_{\boldsymbol{m}_t} \mathbb{E}_{p(\boldsymbol{s}_T|\tilde{s}_{1:t})}[\rho_T(\boldsymbol{m}_t, B_{t,c} - (T-t)||\boldsymbol{m}_t||_0, B_s)] \quad (5)$$

This approach allows us to lift (3d) from (3), to which we add a Lagrange multiplier and achieve:

$$\hat{\rho}_t(\boldsymbol{m}_{t-1}, B_{t,c}, B_s) \triangleq \max_{\boldsymbol{m}_t} \mathbb{E}_{p(\boldsymbol{s}_T|\tilde{s}_{1:t})}[\hat{\rho}_T(\boldsymbol{m}_t, B_s)] + \lambda (B_{t,c} - (T-t)||\boldsymbol{m}_t||_0) \quad (6)$$

for $t = 1, \ldots, T-1$ and $\hat{\rho}_T$ is defined as $\rho_T$ without constraint (3d). Consequently, such a $\hat{\rho}_T$ can be solved in a greedy manner as in (2). Afterwards, we will omit $B_{t,c}$ as a parameter of $\hat{\rho}_T$ as it no longer constrains the solution of $\hat{\rho}_T$. Note that the presence of an *additive* penalty in a *maximization* problem is due to the constraint $B_{T,c} \geq 0 \Leftrightarrow -B_{T,c} \leq 0$ which is typically expected prior to Lagrangian reformulation. The above optimization problem remains NP-hard as $\mathbb{E}_{p(\boldsymbol{s}_T|\tilde{s}_{1:t})}[\hat{\rho}_T(\boldsymbol{m}_t, B_s)]$ is submodular in $\boldsymbol{m}_t$ (see Appendix B). Although greedy approximations exist for submodular optimization, their running time of $O(||\boldsymbol{m}_{t-1}||_0^2)$ remains far too slow due to the large number of network elements in DNNs. Fortunately, it can be significantly simplified by exploiting the following lemma (its proof is in Appendix C.):

**Lemma 1.** *Let $\boldsymbol{e}^{(i)}$ be an $M$-dimensional one-hot vectors with the $i$-th element be 1. $\forall \ 1 \leq a, b \leq M$; $\boldsymbol{m} \in \{0,1\}^M$ s.t. $\boldsymbol{m} \wedge (\boldsymbol{e}^{(a)} \vee \boldsymbol{e}^{(b)}) = 0$. Given a vector of observed saliency $\tilde{s}_{1:t}$, if $\mu_{T|1:t}^a \geq \mu_{T|1:t}^b$ and $\mu_{T|1:t}^a \geq 0$, then*

$$\mathbb{E}_{p(\boldsymbol{s}_T|\tilde{s}_{1:t})}[\rho_T(\boldsymbol{m} \vee \boldsymbol{e}^{(b)})] - \mathbb{E}_{p(\boldsymbol{s}_T|\tilde{s}_{1:t})}[\rho_T(\boldsymbol{m} \vee \boldsymbol{e}^{(a)})] \leq \mu_{T|1:t}^b \Phi(\nu/\theta) + \theta \, \phi(\nu/\theta)$$

*where $\theta \triangleq \sqrt{\sigma_{T|1:t}^{aa} + \sigma_{T|1:t}^{bb} - 2\sigma_{T|1:t}^{ab}}$, $\nu \triangleq \mu_{T|1:t}^b - \mu_{T|1:t}^a$, and $\Phi$ and $\phi$ are standard normal CDF and PDF, respectively.*

Here, '$\vee$' and '$\wedge$' represent bitwise OR and AND operations, respectively. The bitwise OR operation is used to denote the *inclusion* of $\boldsymbol{e}^{(a)}$ or $\boldsymbol{e}^{(b)}$ in $\boldsymbol{m}_t$. Due to the strong tail decay[7] of $\phi$ and $\Phi$, Lemma 1 indicates *at most marginal* possible improvement provided by opting for $\boldsymbol{m}_t = \boldsymbol{m} \vee \boldsymbol{e}^{(b)}$ as opposed to $\boldsymbol{m}_t = \boldsymbol{m} \vee \boldsymbol{e}^{(a)}$ given $\mu_{T|1:t}^a \geq \mu_{T|1:t}^b$.

Lemma 1 admits the following approach to optimize $\hat{\rho}_t$: starting with $\boldsymbol{m}_t = 0^M$, we consider the inclusion of network elements in $\boldsymbol{m}_t$ by the *descending* order of $\{\mu_{T|1:t}^a\}_{a=1}^M$ which can be computed analytically using MOGP. A network element denoted by $\boldsymbol{e}^{(a)}$ is included in $\boldsymbol{m}_t$ if it improves the objective in (5). The algorithm terminates once the highest not-yet-included element does not improve the objective function as a consequence of the penalty term outweighing the improvement in $\mathbb{E}_{p(\boldsymbol{s}_T|\tilde{s}_{1:t})}[\rho_T]$. The remaining excluded elements are then pruned.

Following the algorithm sketch above, we define the utility of network element $v_t^a$ with respect to candidate pruning mask $\boldsymbol{m}_t \dot{\leq} \boldsymbol{m}_{t-1}$ which measures the improvement in $\mathbb{E}_{p(\boldsymbol{s}_T|\tilde{s}_{1:t})}[\rho_T]$ as a

---

[6]We omit (4b) as it is automatically satisfied due to our simplification.

[7]Note as $\mu_{T|1:t}^a \geq \mu_{T|1:t}^b$, $\Phi(\cdot) \leq 0.5$ and experiences tail decay proportional to $\mu_{T|1:t}^a - \mu_{T|1:t}^b$.

consequence of inclusion of $e^{(a)}$ in $m_t$:

$$\Delta(a, m_t, \tilde{s}_{1:t}, B_s) \triangleq \mathbb{E}_{p(s_T|\tilde{s}_{1:t})} \left[ \rho_T(e^{(a)} \vee m_t, B_s) - \rho_T(m_t, B_s) \right]. \tag{7}$$

We can now take a Lagrangian approach to pruning decisions during iteration $t$ by balancing the utility of network element $v_t^a$ against the change of the penalty (i.e., $\lambda(T-t)$) in Algorithm 1. Due to the relatively expensive cost of performing early pruning, we chose to early prune every $T_{step}$ iterations of SGD. Typically $T_{step}$ was chosen to correspond to 10-20 epochs of training. To compute $\Delta(\cdot)$ we sampled from $p(s_T|\tilde{s}_{1:t})$ and used a greedy selection algorithm per sample as in (2). During implementation, we also enforced an additional hard constraint $||m_t||_0 \geq B_s$ which we believe is desirable for practicality reasons. We used a fixed value of $B_{1,c} = ||m_0||_0 T_0 + B_s(T - T_0)$ in all our experiments.

---

**Algorithm 1** Bayesian Early Pruning

---

**Require:** $\mathcal{N}, v_1, T_0, T_{step}, T, B_{1,c}, B_s, \lambda$        ▷ DNN $\mathcal{N}$, Lagrangian penalty $\lambda$
1:   $S_{1:T_0} \leftarrow \text{train}(\mathcal{N}_{v_1}, T_0)$        ▷ Train for $T_0$ iterations to create seed dataset.
2:   $B_{T_0,c} \leftarrow B_{1,c} - T_0 \, dim(v_1)$        ▷ Track computational effort expenditure.
3:   **for** $k \leftarrow 0, \ldots, \frac{T-T_0}{T_{step}}; t \leftarrow T_0 + kT_{step}$ **do**        ▷ Early prune every $T_{step}$ iterations from $T_0$.
4:      $\mu_{T|1:t}, \sigma_{T|1:t} \leftarrow MOGP(S_{1:t})$        ▷ Train and perform inference.
5:      $s_T \leftarrow argsort(-\mu_{T|1:t})$        ▷ Sort descending.
6:      $m_t \leftarrow 0^{dim(v_t)}$        ▷ Initial pruning mask.
7:      **for** $a \leftarrow s_T^1, \ldots, s_T^{dim(v_t)}$ **do**        ▷ Consider each network element.
8:        **if** $B_{t,c} - (T-t)||m_t||_0 > 0$ **then**    ▷ Remaining $B_{t,c}$ budget can support training $v_t^a$.
9:          $m_t = m_t \vee e^{(a)}$
10:       **else if** $\Delta(a, m_t, \tilde{s}_{1:t}, B_s) \geq \lambda(T-t)$ **then**   ▷ Balance utility against change of penalty.
11:         $m_t = m_t \vee e^{(a)}$
12:       **else**
13:         **break**
14:      $prune(v_t, m_t)$        ▷ $dim(v_t)$ is reduced here.
15:      $B_{t+T_{step},c} \leftarrow B_{t,c} - T_{step}||m_t||_0$
16:      $S_{t+1:t+T_{step}} \leftarrow train(\mathcal{N}_{v_t}, T_{step})$        ▷ Continue training with pruned network.
17: **return** $\mathcal{N}$

---

## 4 EXPERIMENTS AND DISCUSSION

We evaluate our modeling approach as well as our BEP algorithm on the CIFAR-10, CIFAR-100 (Krizhevsky, 2009), and ImageNet (Deng et al., 2009) datasets. For CIFAR-10/CIFAR-100 we used a benchmark Convolutional Neural Network (CNN) with 4 convolutional layers, and 1 dense layer.[8] For ImageNet we validated on the ResNet-50 architecture (He et al., 2016a).

Due to the cubic time complexity of MOGPs, we used a variational approximation (Hensman et al., 2015). In all of our models, we used 60 variational inducing points per latent function. We used GPFlow library (Matthews et al., 2017) to build our models.

### 4.1 MODELING EVALUATION

A key assertion in our approach is the importance of capturing co-adaptation and co-evolution effects in network elements. To verify our MOGP approach captures these effects, we compare MOGP vs. GP belief modeling where GP assumes independence in saliency measurements across network elements (i.e., $p(s_{1:T}) \triangleq \prod_{a=1}^{M} p(s_{1:T}^a)$).

A dataset of saliency measurements of convolutional filters and neurons was constructed by instrumenting the training process of our 5-layer CNN on the CIFAR-10/CIFAR-100 dataset. Keras (Chollet, 2015) was used to train this model over 150 epochs.[9]

---

[8]Code available at `https://github.com/keras-team/keras/blob/master/examples/cifar10_cnn.py`
[9]Complete experimental setup details found in Appendix G.2.

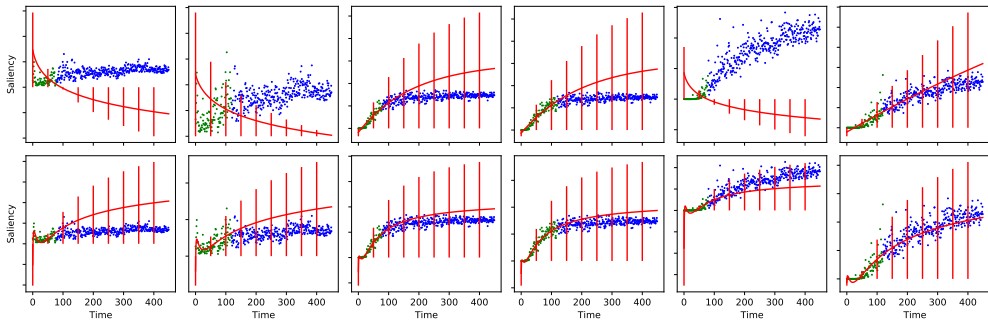

Figure 1: Visualization of qualitative differences between GP and MOGP prediction. Top: GP, Bottom: 18-MOGP. Dataset is separated into training (green) and validation (blue). Posterior belief of the saliency is visualized as predictive mean (red line), and $95\%$ confidence interval (error bar).

Table 1: Comparing log likelihood (standard error) of test data for independent GPs (GP) vs. MOGP with $n$ latent functions ($n$-MOGP) on collected saliency measurements from CIFAR-100 training. Measurements are given as a multiple of $-10^4$ (lower is better). MOGP outperforms GP, particularly on the small dataset. Results are averaged over 20 runs. Extremely large values are due to the GP model being unable to fit the data.

| | Small dataset | | | Medium dataset | | | Large dataset | | |
|---|---|---|---|---|---|---|---|---|---|
| | Lyr 1 | Lyr 2 | Lyr 3 | Lyr 1 | Lyr 2 | Lyr 3 | Lyr 1 | Lyr 2 | Lyr 3 |
| GP | 0.75(0.06) | 5.7(5.7)e4 | 5.6(5.6)e4 | 0.64(0.04) | 0.70(0.04) | **2.13**(**0.05**) | 3.4(3.4)e3 | 0.31(0.02) | 1.06(0.02) |
| 4-MOGP | 0.79(0.05) | 0.98(0.12) | 3.13(0.10) | 0.44(0.04) | 0.60(0.10) | 2.29(0.06) | 0.12(0.01) | 0.24(0.03) | 1.07(0.03) |
| 8-MOGP | 0.65(0.05) | 0.89(0.11) | 3.00(0.09) | 0.38(0.04) | 0.60(0.10) | 2.20(0.06) | 0.10(0.01) | 0.18(0.01) | 1.02(0.03) |
| 18-MOGP | **0.62**(0.05) | **0.84**(0.11) | 2.93(0.10) | 0.36(0.03) | **0.56**(0.10) | 2.22(0.07) | 0.09(0.01) | 0.18(0.01) | 1.01(0.03) |
| 32-MOGP | 0.65(0.05) | 0.85(0.09) | **2.89**(0.10) | **0.36**(0.03) | 0.59(0.10) | 2.16(0.06) | **0.09**(0.02) | **0.18**(0.01) | **1.00**(0.03) |

We trained belief models with small ($t = [0, 26]$ epochs), medium ($t = [0, 40]$ epochs), and large ($t = [0, 75]$ epochs) training dataset of saliency measurements. For GPs, a separate model was trained per network element (convolutional filter, or neuron). For MOGP, all network elements in a single layer[10] shared one MOGP model. We evaluated these models using log likelihood of the remainder of the saliency measurements. We present the performance of the models in Table 1 for CIFAR-100.[11] Our MOGP approach better captures the saliency of network elements than a GP approach. Furthermore, using additional latent functions improves MOGP modeling with diminishing returns. We visualize the qualitative differences between GP and MOGP prediction in Figure 1. We observe that MOGP is able to capture the long term trend of saliency curves with significantly less data than GP.

## 4.2 SMALL-SCALE EXPERIMENTS

We applied the early pruning algorithm on the aforementioned architecture, and training regimen. We investigated the behavior of the penalty parameter, $\lambda$. We observed that the penalty parameter was difficult to tune properly, either being too aggressive at pruning, or too passive. To rectify this issue, we used a feedback loop to determine the penalty at iteration $t$, $\lambda_t$ dynamically. Dynamic penalty scaling[12] uses feedback from earlier pruning iterations to increase or decrease the iteration penalty at time t: $\lambda_t = \lambda\left[(1/\lambda)^{\wedge}\left((T-t)||\boldsymbol{m}_t||_0/B_{t,c} - 1\right)\right]$. The dynamic penalty is increased if the anticipated compute required to complete training, $(T-t)||\boldsymbol{m}_t||_0$ begins to exceed the amount of compute budget remaining, $B_{t,c}$. In such case, a higher penalty is needed to satisfy the computational budget constraint as per (6). We compare dynamic penalty scaling, and penalty without scaling in Fig. 2 using $T_0 = 20$ epochs, $T_{step} = 10$ epochs for the first convolutional layer of our CNN. Going forward, we use dynamic penalty scaling in our experiments.

---

[10]In our observations, jointly modeling the belief of multiple layers' saliency measurements using MOGP yielded no measurable improvement in log-likelihood.

[11]For CIFAR-10 see Appendix G.1.

[12]Further details can be found in Appendix D.

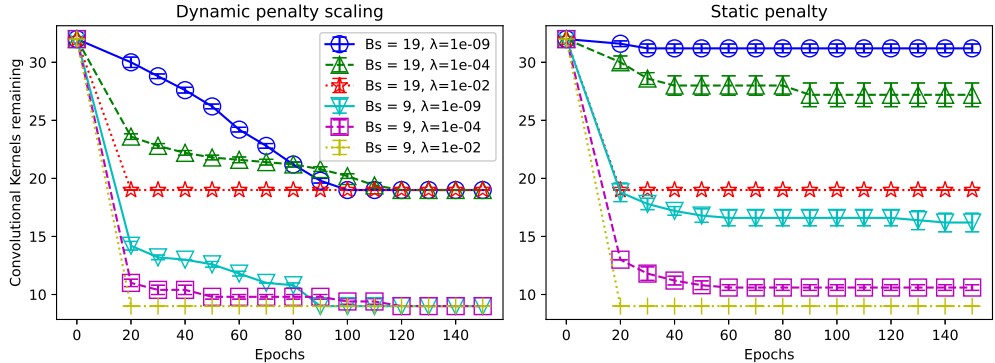

Figure 2: Comparing dynamic penalty scaling vs. static on pruning a 32-convolutional filter layer in a CNN. Dynamic penalty scaling encourages gradual pruning across a wide variety of settings of $\lambda$.

Table 2: Performance (standard error) against percentage of neurons/filters pruned per layer with varying $\lambda$ for tested algorithms.

| | CIFAR-10 | | | | CIFAR-100 | | | |
|---|---|---|---|---|---|---|---|---|
| | 70% | 80% | 90% | 95% | 70% | 80% | 90% | 95% |
| DSR | 74.1(0.2)% | 65.3(0.4)% | 52.8(0.5)% | 22.1(5.0)% | 37.9(0.7)% | 29.7(0.1)% | 17.5(0.3)% | 4.4(1.4)% |
| SNIP | 75.4(4.7)% | 67.7(0.7)% | 50.8(0.8)% | 29.4(4.9)% | 22.9(9.0)% | 15.7(6.1)% | 9.9(3.7)% | 2.2(1.2)% |
| GraSP | 74.6(0.6)% | 66.5(0.9)% | 50.7(0.6)% | 32.9(1.0)% | 28.4(7.0)% | 22.6(5.4)% | 13.9(3.2)% | 1.0(0.0)% |
| BEP 1e−2 | 75.9(0.3)% | 69.7(0.4)% | 54.8(1.0)% | 18.9(5.4)% | 40.6(0.2)% | 32.2(0.6)% | 19.1(0.5)% | 7.1(1.6)% |
| BEP 1e−4 | 75.4(1.7)% | 70.5(3.2)% | 55.7(0.9)% | **36.1(1.1)**% | **41.3(0.3)**% | 32.4(0.3)% | **19.7(0.8)**% | **8.5(0.8)**% |
| BEP 1e−7 | **76.0(0.1)**% | **70.6(0.2)**% | **56.2(0.4)**% | 30.4(5.1)% | 40.6(0.2)% | **33.0(0.5)**% | 19.5(0.5)% | 6.6(1.5)% |

We compare our work with *SNIP* (Lee et al., 2019), *GraSP* (Wang et al., 2020), and momentum-based dynamic sparse remaparameterization (DSR) (Dettmers & Zettlemoyer, 2019). To compare against DSR, we instantiate a smaller network of the size BEP yields after training has completed as it is a prune and regrow method. The SNIP and GraSP approaches are extended to neurons/filters by averaging the saliencies of the constituent weight parameters. We experimented with various degrees of sparsity, using BEP to prune a portion of filters/neurons of each layer.[13] We present the results in Table 2. Our approach better preserves performance at equivalent sparsity. A lower penalty yields higher performing results showing $\lambda$ serves well at balancing performance vs. computational budget.

We investigate the robustness of BEP and MOGP hyperparameters. We vary the number of MOGP variational inducing points, MOGP latent functions, and $T_{step}$ and observe the performance of BEP 1e−4 on CIFAR-10/CIFAR-100 at 80%, 90%, and 95% sparsity. We present these results in Table 3. We observe that in general, all hyperparameters are robust to changes. Mild degradation is observed in the extremal hyperparameter settings.

## 4.3 SPEEDING UP RESNET TRAINING ON IMAGENET

Our chief goal in this work is to speed up training of large-scale DNNs such as ResNet (He et al., 2016a;b) on the ImageNet dataset. Pruning ResNet requires a careful definition of network element saliency to allow pruning of all layers. ResNet contains long sequences of *residual units* with matching number of input/output channels. The inputs of residual units are connected with *shortcut connections* (i.e., through addition) to the output of the residual unit.[14] Due to shortcut connections, this structure requires that within a sequence of residual units, the number of inputs/output channels of all residual units must match exactly. This requires *group pruning* of residual unit channels for

---

[13]In our observations saliency measurements don't well capture network element efficacy when comparing across layers. Thus pruning whole networks using network element saliency yields poor performing networks with bottlenecks. This limitation of saliency functions is well known (See Molchanov et al. (2017) Appendix A.1 and A.2; Wang et al. (2020) Section 3 last paragraph). Development of saliency functions which overcome this shortcoming while remaining performant is a difficult open problem outside the scope of this work.

[14]Precise details of the ResNet architecture may be found in He et al. (2016a) Section 3.

Table 3: Ablation study showing performance (standard error) vs. varying early pruning hyperparameters: MOGP variational inducing points (Ind. pnts.), MOGP latent functions (Lat. func.), $T_{step}$. Default setting for hyperparameters are 60, $1.0\times$, and 10 respectively. Outside of the higest sparsity setting, 95%, all hyperparameters are robust to changes, with mild degradation observed in the extremal settings.

| | | CIFAR-10 | | | CIFAR-100 | | |
|---|---|---|---|---|---|---|---|
| | | 80% | 90% | 95% | 80% | 90% | 95% |
| Ind. pnts. | 26 | 69.6(0.4)% | 56.8(0.2)% | 29.7(4.9)% | 31.7(0.6)% | 19.2(0.5)% | 8.3(0.7)% |
| | 40 | 70.9(0.1)% | 55.6(0.6)% | 30.5(5.1)% | 32.3(0.7)% | 19.6(0.3)% | 6.6(1.4)% |
| | 60 | 70.5(3.2)% | 55.7(0.9)% | 36.1(1.1)% | 32.4(0.3)% | 19.7(0.8)% | 8.5(0.8)% |
| | 90 | 70.4(0.3)% | 55.1(0.7)% | 35.5(1.9)% | 32.6(0.4)% | 18.5(0.6)% | 8.7(0.3)% |
| Lat. func. | $0.25\times$ | 70.4(0.4)% | 55.6(0.8)% | 35.8(0.2)% | 32.6(0.3)% | 16.3(3.8)% | 7.4(1.8)% |
| | $0.50\times$ | 70.0(0.2)% | 56.9(0.4)% | 34.5(0.6)% | 32.1(0.5)% | 18.9(0.7)% | 7.0(1.5)% |
| | $1.0\times$ | 70.5(3.2)% | 55.7(0.9)% | 36.1(1.1)% | 32.4(0.3)% | 19.7(0.8)% | 8.5(0.8)% |
| | $2.0\times$ | 69.8(0.3)% | 55.7(0.7)% | 34.8(0.5)% | 32.0(0.4)% | 20.8(0.2)% | 7.7(0.4)% |
| $T_{step}$ | 2 | 69.2(0.5)% | 54.7(0.6)% | 29.4(5.0)% | 32.1(0.2)% | 20.0(0.3)% | 4.3(1.5)% |
| | 5 | 70.3(0.2)% | 55.6(0.5)% | 31.6(5.4)% | 32.7(0.4)% | 19.4(0.4)% | 5.2(1.8)% |
| | 10 | 70.5(3.2)% | 55.7(0.9)% | 36.1(1.1)% | 32.4(0.3)% | 19.7(0.4)% | 8.5(0.8)% |
| | 20 | 70.3(0.2)% | 56.2(0.1)% | 29.8(5.0)% | 32.8(0.5)% | 19.6(0.4)% | 6.8(1.5)% |

a sequence of residual units, where group pruning an output channel of a residual unit sequence requires pruning it from the inputs/outputs of all residual units within the sequence.[15]

We trained ResNet-50 with BEP as well as SNIP and GraSP.[16] We group pruned less aggressively as residual unit channels feed into a large number of residual units, thus making aggressive pruning likely to degrade performance. We ran BEP iterations at $t = [15, 20, 25, 35, 45, 55, 75]$ epochs. We trained for 100 epochs on $4\times$ Nvidia Geforce GTX 1080 Ti GPUs. More experimental details found in Appendix G.2. We present our results in Table 4. We achieve higher performance than related techniques, albeit at longer wall time. Our approach captures the *training time vs. performance* tradeoff present in DNNs, unlike competing approaches.

## 5 RELATED WORK

**Pruning and related techniques.** Initial works in DNN pruning center around saliency based pruning after training including Skeletonization (Mozer & Smolensky, 1988), Optimal Brain Damage and followup work (Hassibi & Stork, 1992; LeCun et al., 1989) as well as sensitivity based pruning (Karnin, 1990). In recent years, saliency functions been adapted to pruning neurons or convolutional filters. Li et al. (2017) define a saliency function on convolutional *filters* by using the $L_1$ norm. Molchanov et al. (2017) propose using a first-order Taylor-series approximation on the objective function as a saliency measure. Dong et al. (2017) propose layer wise pruning of weight parameters using a Hessian based saliency measure. Several variants of pruning after training exist. Han et al. (2015) propose *iterative pruning* where pruning is performed in stages alternating with fine tune training. Guo et al. (2016) suggest dynamic network surgery, where pruning is performed on-the-fly during evaluation time. Li et al. (2017) and He et al. propose reinforcement learning for pruning decisions. A comprehensive overview may be found in Gale et al. (2019).

Knowledge distillation (Hinton et al., 2015; Lu et al., 2017; Tung & Mori, 2019; Yim et al., 2017) aim to transfer the capabilities of a trained network into a smaller network. Weight sharing (Nowlan & Hinton, 1992; Ullrich et al., 2017) and low rank matrix factorization (Denton et al., 2014; Jaderberg et al., 2014) aim to compress the parameterization of neural networks. Network quantization (Courbariaux et al., 2015; Hubara et al., 2017; Micikevicius et al., 2018) use lower fidelity representation of network elements (e.g. 16-bit) to speed up training and evaluation. Although speedup during training is achievable through network quantization, this technique requires hardware support and only

---

[15]We formally define saliency on residual unit sequences in Appendix G.3

[16]We omit comparison to DSR due to differing underlying deep learning library which makes walltime comparisons inaccurate.

Table 4: BEP vs. SNIP and Grasp on ResNet-50 on ImageNet dataset. We vary percentage of residual unit sequence channels (Seq) and filters pruned (Lyr). 'Train' refers to wall time during network training, 'Prune' refers to pruning modeling/inference overhead. Benchmark wall time for ResNet-50 is 55h on our hardware. Benchmark performance is $75.7\%$ for unpruned ResNet-50.

| | Seq 30%, Lyr 60% | | | | Seq 60%, Lyr 90% | | | | Seq 30%, Lyr 98% | | | |
|---|---|---|---|---|---|---|---|---|---|---|---|---|
| | Top-1 | Top-5 | Train | Prune | Top-1 | Top-5 | Train | Prune | Top-1 | Top-5 | Train | Prune |
| SNIP | 72.0% | 90.6% | 27.9h | 0.7h | 62.0% | 83.8% | 15.8h | 0.7h | 50.9% | 74.2% | 21.7h | 0.7h |
| GraSP | 72.1% | 90.6% | 27.1h | 2.7h | 61.6% | 83.6% | 16.5h | 2.7h | 52.2% | 75.4% | 21.7h | 2.7h |
| BEP $1e-1$ | 72.2% | 90.6% | 31.6h | 2.9h | 62.0% | 83.8% | 22.8h | 1.5h | **53.7%** | 76.8% | 27.8h | 2.5h |
| BEP $1e-4$ | **72.5%** | 91.0% | 34.8h | 2.2h | **62.3%** | 84.5% | 22.0h | 1.8h | 53.5% | 76.6% | 27.8h | 2.6h |

provides coarse granularity in trading off computational effort vs. performance. Current GPUs only extend native support to 16-bit floating point operations. Furthermore, our approach is orthogonal to quantization allowing the techniques to be combined for further speedup.

**Initialization time or training time pruning.** Frankle & Carbin (2019) show that a randomly initialized DNN contains a small subnetwork, which if trained by itself, yields equivalent performance to the original network. SNIP (Lee et al., 2019) and GraSP (Wang et al., 2020) propose pruning connection weights prior to the training process through a first order and second order saliency function respectively. Sparse Evolutionary Training (Mocanu et al., 2018) propose initializing networks with sparse topology prior to training. Narang et al. (2017) consider connection weight pruning during training for recurrent neural networks using a heuristic approach.

Dynamic sparse reparameterization considers pruning and regrowing parameter weights during the training process (Bellec et al., 2018; Dettmers & Zettlemoyer, 2019; Mostafa & Wang, 2019). Dai et al. (2019) propose a grow and prune approach to learning network architecture and connection layout. We differ from existing work as our focus is on speeding up neural network training, meanwhile other works in training time pruning aim to achieve sparse network layouts. To the best of our knowledge, except for small speedups presented in (Dettmers & Zettlemoyer, 2019), the above works do not demonstrate speedup during training time using popular deep learning libraries run on modern GPUs.

PruneTrain (Lym et al., 2019) also proposes pruning filters during training to achieve speedup while minimizing degradation to performance with periodic pruning iterations. In contrast to our approach, PruneTrain does not allow specification of the desired network size after training. A specified network size may be useful if training for resource constrained devices such as mobile phones or edge devices. We compare with PruneTrain under the early pruning problem definition in Appendix E.

## 6 CONCLUSION

This paper presents a novel efficient algorithm to perform pruning of DNN elements such as neurons, or convolutional layers during the training process. To achieve *early* pruning before the training converges while preserving the performance of the DNN upon convergence, a Bayesian model (i.e., MOGP) is used to predict the saliency of DNN elements in the future (unseen) training iterations by exploiting the exponentially decaying behavior of the saliency and the correlations between saliency of different network elements. Then, we exploit a property (Lemma 1) of the objective function and propose an efficient Bayesian early pruning algorithm. Empirical evaluations on benchmark datasets show that our algorithm performs favorably to related works for pruning convolutional filters and neurons. Our approach remains flexible to changes in saliency function, and appropriately *balances* the training time vs. performance tradeoff in training DNNs. We are able to train an early pruned ResNet-50 model achieving a $48.6\%$ speedup (37h vs. 55h) while maintaining a validation accuracy of $72.5\%$.

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

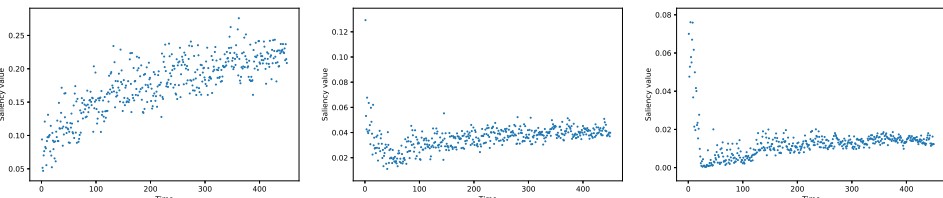

Figure 3: Convolutional filter saliency over 150 epochs of SGD on CIFAR-10.

# A  MODELING DETAILS

## A.1  SALIENCY FUNCTION

In this work, we use a first order Taylor-series saliency function proposed by Molchanov et al. (2017). Our design (Section 3) remains flexible to allow usage of arbitrary saliency functions in a plug-n-play basis. We partition a DNN of $L$ layers, where each layer $\ell$ contains $C_\ell$ convolutional filters, into a sequence of convolutional filters $[z_{\ell,c}]_{\ell=1,\dots,L}^{c=1,\dots,C_\ell}$. Each filter $z_{\ell,c} : \mathbb{R}^{C_{\ell-1} \times W_{\ell-1} \times H_{\ell-1}} \to \mathbb{R}^{W_\ell \times H_\ell}$ can be considered as one *network element* in $\boldsymbol{v}_T$ and $z_{\ell,c}(\mathbf{P}_{\ell-1}) \triangleq \mathcal{R}(\mathbf{W}_{\ell,c} * \mathbf{P}_{\ell-1} + b_{\ell,c})$ where $\mathbf{W}_{\ell,c} \in \mathbb{R}^{C_\ell \times O_\ell \times O'_\ell}$, $b_{\ell,c}$ are kernel weights and bias.with receptive field $O_\ell \times O'_\ell$, '$*$' represents the convolution operation, $\mathcal{R}$ is the activation function, $\mathbf{P}_{\ell-1}$ represents the output of $\boldsymbol{z}_{\ell-1} \triangleq [z_{\ell-1,c'}]_{c'=1,\dots,C_{\ell-1}}$ with $\mathbf{P}_0$ corresponding to an input $\mathbf{x}_d \in \mathcal{X}$, and $W_\ell, H_\ell$ are width and height dimensions of layer $\ell$ for $\ell = 1, \dots, L$. Let $\mathcal{N}_{\boldsymbol{z}_\ell : \boldsymbol{z}_{\ell'}} \triangleq \boldsymbol{z}_{\ell'} \circ, \dots, \circ \boldsymbol{z}_\ell$ denote a *partial* neural network of layers $[\ell, \dots, \ell']_{1 \le \ell \le \ell' \le L}$. The Taylor-series saliency function on the convolutional filter $z_{\ell,c}$ denoted as $s([\ell, c])$ is defined[17]:

$$s([\ell, c]) \triangleq \frac{1}{D} \sum_{d=1}^{D} \left| \frac{1}{W_\ell \times H_\ell} \sum_{j=1}^{W_\ell \times H_\ell} \frac{\partial \mathcal{L}(\mathbf{P}_\ell^{(\mathbf{x}_d)}, y_d; \mathcal{N}_{\boldsymbol{z}_{\ell+1} : \boldsymbol{z}_L})}{\partial P_{\ell,c,j}^{(\mathbf{x}_d)}} P_{\ell,c,j}^{(\mathbf{x}_d)} \right|. \tag{8}$$

where $\mathbf{P}_\ell^{(\mathbf{x}_d)}$ is the output of the partial neural network $\mathcal{N}_{\boldsymbol{z}_1 : \boldsymbol{z}_\ell}$ with $\mathbf{x}_d$ as the input and $[P_{\ell,c,j}^{\mathbf{x}_d}]_{j=1,\dots,W_\ell \times H_\ell}$ interprets the output of the $c$-th filter in vectorized form. This function uses the first-order Taylor-series approximation of $\mathcal{L}$ to approximate the change in loss if $z_{\ell,c}$ was changed to a constant 0 function. Using the above saliency definition, pruning filter $z_{\ell,c}$ corresponds to collectively zeroing $\mathbf{W}_{\ell,c}, b_{\ell,c}$ as well as weight parameters[18] $[\mathbf{W}_{\ell+1,c',\{:,:,c\}}]_{c'=1,\dots,C_{\ell+1}}$ of $\boldsymbol{z}_{\ell+1}$ which utilize the output of $z_{l,c}$. This definition can be extended to elements (e.g. neurons) which output scalars by setting $W_\ell = H_\ell = 1$.

## A.2  ON THE CHOICE OF THE "EXPONENTIAL KERNEL"

We justify our choice of the exponential kernel as a modeling mechanism by presenting visualizations of saliency measurements collected during training, and comparing these to samples drawn from the exponential kernel $k_q(t, t') \triangleq \frac{\beta^\alpha}{(t+t'+\beta)^\alpha}$, as shown in Figs. 3-4. Both the saliency and the function samples exhibit exponentially decaying behavior, which makes the exponential kernel a strong fit for modeling saliency evolution over time.

Furthermore we note that the exponential kernel was used to great effect in Swersky et al. (2014) with respect to modeling loss curves as a function of epochs. Loss curves also exhibit asymptotic behavior, similar to saliency measurement curves, thus providing evidence for the exponential kernel being an apt fit for our task.

## A.3  PREDICTIVE DISTRIBUTION OF THE SALIENCY

Let the *prior* covariance matrix be $\boldsymbol{K}_{\tau_1 : \tau_2} \triangleq [\mathrm{cov}[s_t^a, s_{t'}^{a'}]]_{t,t'=\tau_1,\dots,\tau_2}^{a,a'=1,\dots,M}$ for any $1 \le \tau_1 \le \tau_2 \le T$. Given a vector of observed saliency $\tilde{\mathbf{s}}_{1:t}$, the MOGP regression model can provide a Gaus-

---

[17]For brevity, we omit parameters $\mathcal{X}, \mathcal{Y}, \mathcal{N}_{\boldsymbol{z}_1 : \boldsymbol{z}_L}, \mathcal{L}$.

[18]Here we use $\{\}$ to distinguish indexing into a tensor from indexing into the sequence of tensors $[\mathbf{W}_{\ell+1,c'}]$.

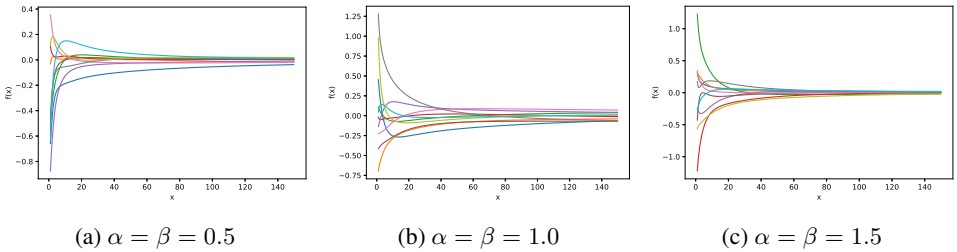

(a) $\alpha = \beta = 0.5$      (b) $\alpha = \beta = 1.0$      (c) $\alpha = \beta = 1.5$

Figure 4: Function samples drawn from the exponential kernel.

sian predictive distribution $p(\boldsymbol{s}_{t'}|\tilde{\mathbf{s}}_{1:t}) = \mathcal{N}(\boldsymbol{\mu}_{t'|1:t}, \boldsymbol{K}_{t'|1:t})$ for any future saliency $\boldsymbol{s}_{t'}$ with the following *posterior* mean vector and covariance matrix: $\boldsymbol{\mu}_{t'|1:t} \triangleq \boldsymbol{K}_{[t't]}\boldsymbol{K}_{1:t}^{-1}\tilde{\mathbf{s}}_{1:t}$, $\boldsymbol{K}_{t'|1:t} \triangleq \boldsymbol{K}_{t':t'} - \boldsymbol{K}_{[t't]}\boldsymbol{K}_{1:t}^{-1}\boldsymbol{K}_{[t't]}^{\top}$ where $\boldsymbol{K}_{[t't]} \triangleq [\mathrm{cov}[s_{t'}^{a}, s_{\tau}^{a'}]]_{\tau=1,\dots,t}^{a,a'=1,\dots,M}$. Then, the $a$-th element $\mu_{t'|1:t}^{a}$ of $\boldsymbol{\mu}_{t'|1:t}$ is the predictive mean of the saliency $s_{t'}^{a}$. And the $[a, a']$-th element of $\boldsymbol{K}_{[t't]}$ denoted as $\sigma_{t'|1:t}^{aa'}$ is the predictive (co)variance between the saliency $s_{t'}^{a}$ and $s_{t'}^{a'}$.

# B   SUBMODULARITY OF $\mathbb{E}[\hat{\rho}_T]$

In (6), the problem of choosing $\boldsymbol{m}$ from $\{0, 1\}^M$ can be considered as selecting a subset $A$ of indexes from $\{1, \dots, M\}$ such that $m_t^a = 1$ for $a \in A$, and $m_t^a = 0$ otherwise. Therefore, $P(\boldsymbol{m}) \triangleq \mathbb{E}_{p(\boldsymbol{s}_T|\tilde{\boldsymbol{s}}_{1:t})}[\hat{\rho}_T(\boldsymbol{m}, B_s)]$ can be considered as a set function which we will show to be submodular. To keep notation consistency, we will remain using $P(\boldsymbol{m})$ instead of representing it as a function of the index subset $A$.

**Lemma 2 (Submodularity).** *Let $\boldsymbol{m}'$, $\boldsymbol{m}'' \in \{0, 1\}^M$, and $e^{(a)}$ be arbitrary M-dimensional one hot vector with $1 \le a \le M$. We have $P(\boldsymbol{m}' \vee e^{(a)}) - P(\boldsymbol{m}') \ge P(\boldsymbol{m}'' \vee e^{(a)}) - P(\boldsymbol{m}'')$ for any $\boldsymbol{m}' \dot{\le} \boldsymbol{m}''$, $\boldsymbol{m}' \wedge e^{(a)} = 0$, and $\boldsymbol{m}'' \wedge e^{(a)} = 0$.*

*Proof.* According to (3),

$$\mathbb{E}_{p(\boldsymbol{s}_T|\tilde{\boldsymbol{s}}_{1:t})}[\hat{\rho}_T(\boldsymbol{m}, B_s)] = \mathbb{E}_{p(\boldsymbol{s}_T|\tilde{\boldsymbol{s}}_{1:t})}\left[\max_{\boldsymbol{m}_T}\left[\boldsymbol{m}_T \cdot \tilde{\boldsymbol{s}}_T, \text{ s.t. } ||\boldsymbol{m}_T||_0 \le B_s, \boldsymbol{m}_T \dot{\le} \boldsymbol{m}\right]\right]$$

Let $\alpha(\boldsymbol{m}) \triangleq \arg\max_{\boldsymbol{m}_T}\left[\boldsymbol{m}_T \cdot \tilde{\boldsymbol{s}}_T, \text{ s.t. } ||\boldsymbol{m}_T||_0 \le B_s, \boldsymbol{m}_T\dot{\le}\boldsymbol{m}\right]$ return the optimized mask $\boldsymbol{m}_T$ given any $\boldsymbol{m}$, $\Lambda_{\boldsymbol{m}} \triangleq \min(\alpha(\boldsymbol{m}) \odot \boldsymbol{s}_T)$ be the minimal saliency of the network elements selected at iteration $T$ for $P(\boldsymbol{m})$. Then, we have

$$P(\boldsymbol{m} \vee e^{(a)}) = \mathbb{E}_{p(\boldsymbol{s}_T|\tilde{\boldsymbol{s}}_{1:t})}\left[\hat{\rho}_T(\boldsymbol{m} \vee e^{(a)}, B_s)\right]$$
$$= \mathbb{E}_{p(\boldsymbol{s}_T|\tilde{\boldsymbol{s}}_{1:t})}\left[\hat{\rho}_T(\boldsymbol{m}, B_s) - \Lambda_{\boldsymbol{m}} + \max(s_T^a, \Lambda_{\boldsymbol{m}})\right]$$

The second equality is due to the fact that the network element $v_T^a$ would only replace the lowest included element in $\boldsymbol{m}_T$ in order to maximize the objective. Then,

$$P(\boldsymbol{m} \vee e^{(a)}) - P(\boldsymbol{m})$$
$$= \mathbb{E}_{p(\boldsymbol{s}_T|\tilde{\boldsymbol{s}}_{1:t})}\left[\hat{\rho}_T(\boldsymbol{m}, B_s) - \Lambda_{\boldsymbol{m}} + \max(s_T^a, \Lambda_{\boldsymbol{m}})\right] - \mathbb{E}_{p(\boldsymbol{s}_T|\tilde{\boldsymbol{s}}_{1:t})}\left[\hat{\rho}_T(\boldsymbol{m}, B_s)\right]$$
$$= \mathbb{E}_{p(\boldsymbol{s}_T|\tilde{\boldsymbol{s}}_{1:t})}\left[-\Lambda_{\boldsymbol{m}} + \max(s_T^a, \Lambda_{\boldsymbol{m}})\right]$$
$$= \mathbb{E}_{p(\boldsymbol{s}_T|\tilde{\boldsymbol{s}}_{1:t})}\left[\max(s_T^a - \Lambda_{\boldsymbol{m}}, 0)\right] \tag{9}$$

Given $\boldsymbol{m}' \dot{\le} \boldsymbol{m}''$, we have $\Lambda_{\boldsymbol{m}'} \le \Lambda_{\boldsymbol{m}''}$ since $\boldsymbol{m}_T \dot{\le} \boldsymbol{m}$ in $\alpha(\boldsymbol{m}')$ is a tighter constraint than that in $\alpha(\boldsymbol{m}'')$. Consequently, we can get $s_t^a - \Lambda_{\boldsymbol{m}'} \ge s_t^a - \Lambda_{\boldsymbol{m}''}$, and thus

$$[P(\boldsymbol{m}' \vee e^{(a)}) - P(\boldsymbol{m}')] \ge [P(\boldsymbol{m}'' \vee e^{(a)}) - P(\boldsymbol{m}'')].$$

$\square$

## C PROOF OF LEMMA 1

We restate Lemma 1 for clarity.

**Lemma 1.** *Let $e^{(i)}$ be an $M$-dimensional one-hot vectors with the $i$-th element be 1. $\forall\ 1 \leq a, b \leq M;\ \boldsymbol{m} \in \{0,1\}^M\ s.t.\ \boldsymbol{m} \wedge (e^{(a)} \vee e^{(b)}) = 0$. Given a vector of observed saliency $\tilde{\mathbf{s}}_{1:t}$, if $\mu^a_{T|1:t} \geq \mu^b_{T|1:t}$ and $\mu^a_{T|1:t} \geq 0$, then*

$$\mathbb{E}_{p(\boldsymbol{s}_T|\tilde{\mathbf{s}}_{1:t})}[\rho_T(\boldsymbol{m} \vee e^{(b)})] - \mathbb{E}_{p(\boldsymbol{s}_T|\tilde{\mathbf{s}}_{1:t})}[\rho_T(\boldsymbol{m} \vee e^{(a)})] \leq \mu^b_{T|1:t}\,\Phi(\nu/\theta) + \theta\,\phi(\nu/\theta)$$

*where $\theta \triangleq \sqrt{\sigma^{aa}_{T|1:t} + \sigma^{bb}_{T|1:t} - 2\sigma^{ab}_{T|1:t}}$, $\nu \triangleq \mu^b_{T|1:t} - \mu^a_{T|1:t}$, and $\Phi$ and $\phi$ are standard normal CDF and PDF, respectively.*

To prove this Lemma, we prove the following first:

**Lemma 3.** $\mathbb{E}_{p(\boldsymbol{s}_T|\tilde{\mathbf{s}}_{1:t})}\left[\rho_T(\boldsymbol{m} \vee e^{(b)})\right] - \mathbb{E}_{p(\boldsymbol{s}_T|\tilde{\mathbf{s}}_{1:t})}\left[\rho_T(\boldsymbol{m} \vee e^{(a)})\right] \leq \mathbb{E}[\max(s^b_T - s^a_T, 0)].$

*Proof.* Due to (9), we have

$$\mathbb{E}_{p(\boldsymbol{s}_T|\tilde{\mathbf{s}}_{1:t})}\left[\rho_T(\boldsymbol{m} \vee e^{(b)})\right] - \mathbb{E}_{p(\boldsymbol{s}_T|\tilde{\mathbf{s}}_{1:t})}\left[\rho_T(\boldsymbol{m} \vee e^{(a)})\right]$$

$$= P(\boldsymbol{m} \vee e^{(b)}) - P(\boldsymbol{m}) - (P(\boldsymbol{m} \vee e^{(a)}) - P(\boldsymbol{m}))$$

$$= \mathbb{E}_{p(\boldsymbol{s}_T|\tilde{\mathbf{s}}_{1:t})}\left[\max(s^b_T - \Lambda_{\boldsymbol{m}}, 0)\right] - \mathbb{E}_{p(\boldsymbol{s}_T|\tilde{\mathbf{s}}_{1:t})}\left[\max(s^a_T - \Lambda_{\boldsymbol{m}}, 0)\right]$$

$$= \mathbb{E}_{p(\boldsymbol{s}_T|\tilde{\mathbf{s}}_{1:t})}\left[\max(s^b_T - \Lambda_{\boldsymbol{m}}, 0) - \max(s^a_T - \Lambda_{\boldsymbol{m}}, 0)\right] \tag{10}$$

$$= \mathbb{E}_{p(\boldsymbol{s}_T|\tilde{\mathbf{s}}_{1:t})}\left[\max(s^b_T - s^a_T, \Lambda_{\boldsymbol{m}} - s^a_T) - \max(0, \Lambda_{\boldsymbol{m}} - s^a_T)\right] \tag{11}$$

$$\leq \mathbb{E}_{p(\boldsymbol{s}_T|\tilde{\mathbf{s}}_{1:t})}\left[\max(s^b_T - s^a_T, 0)\right] \tag{12}$$

The equality (11) is achieved by adding $\Lambda_{\boldsymbol{m}} - s^a_T$ in each term of the two max functions in (10). The inequality (12) can be proved by considering the following two cases:

If $\Lambda_{\boldsymbol{m}} - s^a_T \geq 0$, then

$$\max(s^b_T - s^a_T, \Lambda_{\boldsymbol{m}} - s^a_T) - \max(0, \Lambda_{\boldsymbol{m}} - s^a_T)$$
$$= \max(s^b_T - s^a_T, \Lambda_{\boldsymbol{m}} - s^a_T) - (\Lambda_{\boldsymbol{m}} - s^a_T)$$
$$= \max(s^b_T - s^a_T - (\Lambda_{\boldsymbol{m}} - s^a_T), 0)$$
$$\leq \max(s^b_T - s^a_T, 0)\,.$$

If $\Lambda_{\boldsymbol{m}} - s^a_T < 0$, then

$$\max(s^b_T - s^a_T, \Lambda_{\boldsymbol{m}} - s^a_T) - \max(0, \Lambda_{\boldsymbol{m}} - s^a_T)$$
$$= \max(s^b_T - s^a_T, \Lambda_{\boldsymbol{m}} - s^a_T)$$
$$\leq \max(s^b_T - s^a_T, 0)\,.$$

$\square$

Next we utilize a well known bound regarding the maximum of two Gaussian random variables (Nadarajah & Kotz, 2008), which we restate:

**Lemma 4.** *Let $s^a, s^b$ be Gaussian random variables with means $\mu^a, \mu^b$ and standard deviations $\sigma^a, \sigma^b$, then $\mathbb{E}[\max(s^a, s^b)] \leq \mu^a \Phi\left(\frac{\mu^b - \mu^a}{\theta}\right) + \mu^b \Phi\left(\frac{\mu^b - \mu^a}{\theta}\right) + \theta\phi\left(\frac{\mu^b - \mu^a}{\theta}\right)$ where $\theta \triangleq \sqrt{[\sigma^b]^2 + [\sigma^a]^2 - 2cov(s^b, s^a)}$ and $\Phi, \phi$ are standard normal CDF and PDF respectively.*

Then,

$$
\begin{aligned}
&\mathbb{E}_{p(\boldsymbol{s}_T|\tilde{\mathbf{s}}_{1:t})}[\max(s_T^b - s_T^a, 0)] \\
&= \mathbb{E}_{p(\boldsymbol{s}_T|\tilde{\mathbf{s}}_{1:t})}[\max(s_T^b, s_T^a)] - \mathbb{E}_{p(\boldsymbol{s}_T|\tilde{\mathbf{s}}_{1:t})}[s_T^a] \\
&\leq (\mu_{T|1:t}^b + \mu_{T|1:t}^a)\Phi\big(\frac{\mu_{T|1:t}^b - \mu_{T|1:t}^a}{\theta}\big) + \theta\phi\big(\frac{\mu_{T|1:t}^b \mu_{T|1:t}^a}{\theta}\big) - \mu_{T|1:t}^a \\
&= \mu_{T|1:t}^b\Phi\big(\frac{\mu_{T|1:t}^b - \mu_{T|1:t}^a}{\theta}\big) + \theta\phi\big(\frac{\mu_{T|1:t}^b \mu_{T|1:t}^a}{\theta}\big) + \mu_{T|1:t}^a\left(\Phi\left(\frac{\mu_{T|1:t}^b - \mu_{T|1:t}^a}{\theta}\right) - 1\right) \\
&\leq \mu_{T|1:t}^b\Phi\big(\frac{\mu_{T|1:t}^b - \mu_{T|1:t}^a}{\theta}\big) + \theta\phi\big(\frac{\mu_{T|1:t}^b - \mu_{T|1:t}^a}{\theta}\big)
\end{aligned}
$$

The first inequaltiy follows from Lemma 4. The second inequaltiy is due to $\Phi\big(\frac{\mu_{T|1:t}^b - \mu_{T|1:t}^a}{\theta}\big) \leq 1$ and $\mu_{T|1:t}^a \geq 0$.

## D    DYNAMIC PENALTY SCALING AS A FEEDBACK LOOP

We designed a feedback loop to automatically determine $\lambda_t$ during early pruning. A proportional feedback loop can be defined as follows[19]:

$$\lambda_t \triangleq \lambda + K_p \times e(t) \tag{13}$$

where $K_p \geq 0$ is a proportional constant which modulates $\lambda_t$ according to a signed measure of error $e(\cdot)$ at time $t$. Note that $\lambda_t \geq \lambda$ as $e(t) \geq 0$, and the opposite occurs if $e(t) \leq 0$, which allows the *error* to serve as *feedback* to determine $\lambda_t$. Implicitly, $\lambda_t$ asserts some control over $e(t+1)$, and thus closing the feedback loop.

Traditional PID approaches to determine $K_p$ do not work in our case as $\lambda$ may vary over several orders of magnitude. Consequently, a natural choice for $K_p$ is $\lambda$ itself which preserves the same order of magnitude between $K_p$ and $\lambda$:

$$\lambda_t = \lambda + \lambda \times e(t) = \lambda(1 + e(t)). \tag{14}$$

Here we make two decisions to adapt the above to our task. First, as $\lambda$ is likely to be extremely small, we use exponentiation, as opposed to multiplication. Secondly as $\lambda \leq 1$ in practice, we use $1 - e(t)$ as an exponent:

$$\lambda_t = \lambda^{\wedge}[1 - e(t)] = \lambda\left[(1/\lambda)^{\wedge}e(t)\right]. \tag{15}$$

The above derivation is complete with our definition of $e(t)$:

$$e(t) \triangleq (T-t)\|\boldsymbol{m}_t\|_0/B_{t,c} - 1. \tag{16}$$

The above determines error by the discrepancy between the anticipated compute required to complete training $(T-t)\|\boldsymbol{m}_t\|_0$, vs. the remaining budget $B_{t,c}$ with $e(t) = 0$ if the two are equal. This is a natural measure of feedback for $\lambda$ as we expect the two to be equal if $\lambda$ is serving well to *early prune* the network.

## E    COMPARISON WITH PRUNETRAIN

We compare with PruneTrain (Lym et al., 2019) in Table 5. PruneTrain uses an orthogonal technique of dynamically increasing the minibatch size to achieve further wall time improvements. This prevents accurate wall time comparisons between BEP and PruneTrain. To compare with PruneTrain which

---

[19]This approach is inspired from Proportional-Integral-Derivative (PID) controllers (Bellman, 2015), see Åström et al. (1993) for an introductory survey.

Table 5: Comparing BEP with PruneTrain on ResNet-50 on ImageNet dataset. PruneTrain uses a stronger ResNet-50 baseline with Top-1 76.2% performance. BEP uses a 75.7% baseline. 'Train' refers to wall time during network training, 'Prune' refers to pruning modeling/inference overhead. Benchmark wall time for ResNet-50 is 55h on our hardware. Comparison performed using 47% of ResNet-50 baseline inference FLOPs. Train FLOPs refers to proportion of ResNet-50 baseline training FLOPs used to train the network.

| | 47% Inference FLOPs | | | | |
|---|---|---|---|---|---|
| | Top-1 ($\Delta$) | Top-5 | Train | Prune | Train FLOPs |
| PruneTrain | 74.3% ($-1.9\%$) | - | - | - | 60% |
| BEP 1e$-$1 | 74.1% ($-1.6\%$) | 91.8% | 35.5h | 3.9h | 55.2% |
| BEP 1e$-$4 | 73.9% ($-1.8\%$) | 91.7% | 36.5h | 3.2h | 56.0% |

Table 6: Notations used elsewhere in the paper.

| Notation | Definition |
|---|---|
| $M$ | Total number of network elements in Neural Network. |
| $T$ | Total iterations of SGD in training procedure. |
| $s_t^a$ | Random variable representing the saliency measurement of network element $a$ at time $t$. |
| $\boldsymbol{s}_t$ | Sequence of random variables $[s_t^a]_{a=1,...,M}$ |
| $\boldsymbol{s}_{\tau_1:\tau_2}$ | Sequence of random variables $[\boldsymbol{s}_t]_{t=1...,T}$. |
| $\tilde{\mathbf{s}}_{1:t}$ | The realization of random variable $\boldsymbol{s}_{1:t}$. |
| $\boldsymbol{m}_t$ | Pruning mask at time $t$ |
| $B_s$ | Trained network sparsity budget. |
| $B_{t,c}$ | Computational effort budget at time $t$. |
| $\mu_{t'|1:t}^a$ | $\mathbb{E}[s_{t'}^a \mid \tilde{\mathbf{s}}_{1:t}]$ |
| $\sigma_{t'|1:t}^{aa'}$ | $\text{cov}[s_{t'}^a, s_{t'}^{a'} \mid \tilde{\mathbf{s}}_{1:t}]$ |

does not constrain the trained network size (3b), we train ResNet-50 under equivalent inference cost as a network trained by PruneTrain. To compute train/inference cost (FLOPs) for a convolutional layer, we used a formula defined in Molchanov et al. (2017) A.1:

$$\text{FLOPs} \triangleq 2HW(C_{in}K^2 + 1)C_{out} \tag{17}$$

where $H$, $W$, $C_{in}$ are input height, width, and channels respectively, $K$ is the convolutional kernel size, and $C_{out}$ is the number of output channels of the layer.

Under equivalent inference cost, BEP 1e$-$1 outperforms PruneTrain in Top-1 performance. We also find that BEP 1e$-$1 and BEP 1e$-$4 consumes fewer training FLOPs when compared to baseline.

It should be noted that PruneTrain does not provide a mechanism to constrain the trained network size, thus it is unclear how to utilize it in order to solve the early pruning problem (3), (4).

## F    TABLE OF NOTATIONS

We list a table of notations used elsewhere in the paper in Table 6.

Table 7: Comparing log likelihood (standard error) of test data for Independent GPs (GP) vs. MOGP with n latent functions (n-MOGP) on collected saliency measurements from CIFAR-10 training.

| | Small dataset | | | Medium dataset | | | Large dataset | | |
|---|---|---|---|---|---|---|---|---|---|
| | Lyr 1 | Lyr 2 | Lyr 3 | Lyr 1 | Lyr 2 | Lyr 3 | Lyr 1 | Lyr 2 | Lyr 3 |
| GP | 1.19(0.5) | 1.08(0.06) | 1.07(1.07)e5 | 0.96(0.04) | 0.93(0.03) | 2.47(0.04) | 0.49(0.01) | 0.48(0.01) | 1.33(0.02) |
| 4-MOGP | 1.15(0.05) | 0.89(0.06) | 2.44(0.05) | 0.91(0.02) | 0.80(0.03) | 2.20(0.03) | 0.38(0.02) | 0.39(0.02) | 1.25(0.02) |
| 8-MOGP | 1.09(0.04) | 0.86(0.05) | 2.38(0.04) | 0.84(0.03) | 0.78(0.03) | 2.16(0.03) | 0.32(0.01) | 0.35(0.02) | 1.20(0.02) |
| 18-MOGP | 0.97(0.04) | **0.80**(0.05) | 2.33(0.04) | 0.89(0.03) | 0.76(0.03) | 2.13(0.03) | 0.31(0.01) | 0.35(0.02) | 1.20(0.02) |
| 32-MOGP | **0.96**(0.06) | 0.81(0.06) | **2.32**(0.04) | **0.79**(0.03) | **0.74**(0.03) | **2.13**(0.03) | **0.31**(0.01) | **0.34**(0.02) | **1.20**(0.02) |

## G    MORE EXPERIMENTAL RESULTS AND EXPERIMENTAL DETAILS

### G.1    GP VS. MOGP LOG-LIKELIHOOD ON CIFAR-10 DATASET

Table 7 presents the results of the experiment in Section 4.1 for the CIFAR-10 dataset.

### G.2    EXPERIMENTAL DETAILS

To train our CIFAR-10 and CIFAR-100 models we used an Adam optimizer (Kingma & Ba, 2015) with an initial learning rate of $0.001$. The learning rate used an exponential decay of $k = 0.985$, and a batch size of 32 was used. Training was paused three times evenly spaced per epoch. During this pause, we collected saliency measurements using $40\%$ of the training dataset. This instrumentation subset was randomly select from the training dataset at initialization, and remained constant throughout the training procedure. We performed data preprocessing of saliency evaluations into a standardized $[0, 10]$ range.[20] We used (8) to measure saliency of neurons/convolutional filters. For the convolutional layers we used 12 latent MOGP functions. For the dense layer we used 4 latent MOGP functions.

For our ResNet-50 model we used an SGD with Momentum optimizer with an initial learning rate of $0.1$. The learning rate was divided by ten at $t = [30, 60, 80]$ epochs. We collected saliency data every 5 iterations of SGD, and averaged them into buckets corresponding to 625 iterations of SGD to form our dataset. We used a minimum of 4 latent functions per MOGP, however this was dynamically increased if the model couldn't fit the data up to a maximum of 15.

We sampled 10K points from our MOGP model to estimate $\Delta(\cdot)$ for CIFAR-10/CIFAR-100. For ResNet we sampled 15K points. We repeated experiments 5 times for reporting accuracy on CIFAR-10/CIFAR-100.

### G.3    PRUNING ON RESNET

ResNet architecture is composed of a sequence of residual units: $Z_\ell \triangleq \mathcal{F}(\mathbf{P}_{\ell-1}) + \mathbf{P}_{\ell-1}$, where $\mathbf{P}_{\ell-1}$ is the output of the previous residual unit $Z_{\ell-1}$ and '+' denotes elementwise addition. Internally, $\mathcal{F}$ is typically implemented as three stacked convolutional layers: $\mathcal{F}(\mathbf{P}_{\ell-1}) \triangleq [z_{\ell_3} \circ z_{\ell_2} \circ z_{\ell_1}](\mathbf{P}_{\ell-1})$ where $z_{\ell_1}, z_{\ell_2}, z_{\ell_3}$ are convolutional layers. Within this setting we consider convolutional filter pruning. Although $z_{\ell_1}, z_{\ell_2}$ may be pruned using the procedure described earlier. Pruning $z_{\ell_3}$ requires a different procedure. Due to the direct addition of $\mathbf{P}_{\ell-1}$ to $\mathcal{F}(\mathbf{P}_{\ell-1})$, the output dimensions of $Z_{\ell-1}$ and $z_{\ell_3}$ must match exactly. Thus a ResNet architecture consists of sequences of residual units of length $B$ with matching input/output dimensions: $\zeta \triangleq [Z_\ell]_{\ell=1,\ldots,B}$, s.t. $dim(\mathbf{P}_1) = dim(\mathbf{P}_2) = \ldots = dim(\mathbf{P}_B)$. We propose *group pruning* of layers $[z_{\ell_3}]_{\ell=1,\ldots,B}$ where filters are removed from all $z_{\ell_3}$ in a residual unit sequence in tandem. We define $\boldsymbol{s}([\zeta, c]) \triangleq \sum_{\ell=1}^{B} s([\ell_3, c])$, where $s(\cdot)$ is defined for convolutional layers as in (8). To prune the channel $c$ from $\zeta$, we prune it from each layer in $[z_{\ell 3}]_{\ell=1,\ldots,B}$. Typically we pruned sequence channels less aggressively than convolutional filters as these channels feed into several convolutional layers.

---

[20]Generally, saliency evaluations are relatively small ($\leq 0.01$), which leads to poor fitting models or positive log-likelihood. Precise details of our data preprocessing is in Appendix G.4.

### G.4 DATA PREPROCESSING

We followed the same data preprocessing procedure for both our small scale and ImageNet experiments. To standardize the saliency measurements for a training dataset $\tilde{s}_{1:t}$ in our modeling experiments we clip them between $0$ and an upper bound computed as follows: $ub \triangleq percentile(\tilde{s}_{1:t}, 95) \times 1.3$. This procedure removes outliers. We used $1.3$ as a multiplier, as this upper bound is used to transform test dataset as well, which may have higher saliency evaluations.

After clipping the training data, we perform a trend check for each element $v^a$ by fitting a Linear Regression model to the data $\tilde{s}_{1:t}^a$. For $\tilde{s}_{1:t}^a$ with an increasing trend (i.e., the linear regression model has positive slope) we perform the transformation $\tilde{s}_{1:t}^a = ub - \tilde{s}_{1:t}^a$. The reasoning behind this is that the exponential kernel strongly prefers *decaying* curves. After this preprocessing, we scale up the saliency measurements to a $[0, 10]$ range: $\tilde{s}_{1:t} = \tilde{s}_{1:t} \times 10$. We found that without scaling to larger values, log-likelihood of our models demonstrated extremely high positive values due to small values of unscaled saliency measurements.

We transform the test data in our modeling experiments $\tilde{s}_{t+1:T}$ with the same procedure using the same $ub$ and per-element $v^a$ regression models as computed by the training data. We measure log-likelihood after this transformation for both the test dataset in our small scale experiments.

During the BEP Algorithm, the same steps are followed, however we inverse the trend check transformation ($\tilde{s}_{1:t}^a = ub - \tilde{s}_{1:t}^a$) on the predicted MOGP distribution of $s_T$ prior to sampling for estimation of $\Delta(\cdot)$.

