# OpenReview forum: "Balancing training time vs. performance with Bayesian Early Pruning"
_ICLR.cc/2021/Conference — Reject_

### Official Review · AnonReviewer1 · 2020-10-26
**A complex approach without significant improvement**

**Rating:** 5
**Confidence:** 5

**Review:**

The paper proposes a Bayesian-based approach to early prune parameters, which are predicted to have low saliency/importance, with the goal of accelerating the training of deep neural networks. The predictor is a "multi-output Gaussian process" which is computation expensive.

The writing quality and clarity of this paper is OK, but I recommend the authors to include a table of mathematical notations.

The idea of using a Gaussian process based predictor to predict the importance of parameters during the training process is interesting. However, this paper only compares with some methods (e.g. SNIP and GraSP) which prune parameters before training, inspired by the lottery ticket hypothesis paper. Lots of pruning methods can prune parameters during training before the lottery ticket hypothesis paper appears. For example, PruneTrain https://arxiv.org/abs/1901.09290 can reduce end-to-end training time of ResNet-50 by 39% without losing accuracy by simply using a previous sparsity regularization method (while this paper has a significant accuracy drop to 72.5% for ResNet-50). The paper should compare with those more superior methods, regardless of the fact that in Table 3 it is unclear if BEP can outperform SNIP/GraSP or not under the same "Time".

Moreover, the method introduces new hyperparameters. To tune the hyperparameters, the method should be run multiple times. I cannot see how it will make training a neural network faster, unless the hyperparameters are super robust to generate, which is not deeply discussed.

---

> ### Author Response · Authors · 2020-11-20
> **Review responses**
>
> Thanks for your suggestions on the notations, related work, and hyperparameters. We have discussed the robustness of hyperparameters under the common concerns and will address your other concerns below:
>
> Q1. I recommend the authors to include a table of mathematical notations.
>
> We have provided a Table of Notations (Table 6, Appendix F) in our new draft.
>
> Q2. Lots of pruning methods can prune parameters during training before the lottery ticket hypothesis paper appears.
>
> We agree a number of works have considered some form of pruning during training. We have discussed some (but certainly not all) of these works in the Related Work section, including (Narang et. al. 2017, Mocanu et. al. 2018), as well as several variants of dynamic sparse reparameterization including DEEP-R (Bellec et. al. 2018). To the best of our knowledge, these approaches do not generally result in wall time improvement and often require increased training time in order to train networks which allow for efficient inference *after* training. The only exception to this which we were aware of was the work of Dettmers and Zettlemoyer (2019), which shows only a modest (32%) speedup even with a very high percentage (95%) of connection weights being pruned (Table 3 [2]).
>
> We conjecture that these works do not show practical speedup as they focus on connection pruning during training which yields sparse weight matrices. Sparse matrix operations cannot easily leverage parallel architecture of GPUs (footnote 2).
>
> Q3. PruneTrain can reduce end-to-end training time of ResNet-50 by 39% without losing accuracy by simply using a previous sparsity regularization method (while this paper has a significant accuracy drop to 72.5% for ResNet-50).
>
> We sincerely apologize for missing a related work (due to its publication in a systems conference, it was outside our field of vision), and thank the reviewer for highlighting it for us. We have added a comparison to this work in Table 5 in Appendix E. To allow for a fair comparison, we trained a BEP network of equivalent inference cost (47% inference FLOPs) as a PruneTrain pruned network. In this setting, BEP 1e-1 and BEP 1e-4 outperform PruneTrain by 0.3% and 0.1%, respectively.
>
> We would like to point out that our reported figure of 72.5% (-3.2%) consumes only 28% of the baseline FLOPs at inference while PruneTrain (-1.87%) consumes 47% (Table 1 in [1]) of baseline inference FLOPs.
>
> Although we would like to do further comparisons with PruneTrain (e.g., with higher portion of pruned filters), the unavailability of the code makes this difficult. We think it is hard to reimplement their approach in the limited discussion time available. We certainly will endeavor to include further comparisons to PruneTrain in a future draft.
>
> To achieve further improvement in wall time which yields the 39% improvement, PruneTrain proposes an orthogonal technique to our approach where the minibatch size is dynamically increased as a structurally pruned network requires lower memory on GPUs. We assert that this technique is orthogonal to ours and can be combined with our approach as well as with SNIP/GraSP to achieve further speedup.
>
> Q4. In Table 3 it is unclear if BEP can outperform SNIP/GraSP or not under the same "Time"
>
> We do not claim that BEP can outperform SNIP/GraSP under the same time. We would like to highlight that BEP captures the *balance/tradeoff* between performance (after training) and time which is not addressed by SNIP/GraSP which solely prune during initialization. In our experiments, we controlled the size of the networks after training (i.e., same inference time). Although it is possible to control for the training time, this would make SNIP/GraSP/BEP to have pruned models with different size, and thus different inference time, which does not appropriately solve the early pruning problem (constraint 3b, $\lvert\lvert m_T \rvert\rvert_0 \leq B_s$). Constraint 3b is important as often training is performed to yield a network of a specific size for usage in resource constrained scenarios (e.g. mobile phones). We have mentioned this use case and importance of constraint 3b in our problem statement in our new draft.
>
> Q5: Improvements over SNIP/GraSP.
>
> We highlight that on CIFAR-10/CIFAR-100, our BEP outperforms competing approaches by a significant margin. In our new draft, we have also added a new experiment which prunes a much larger portion of convolutional filters in ResNet-50.\. In this setting, BEP 1e-1 outperforms SNIP by 2.8\%, and GraSP by 1.5\% in Top-1 performance. This may be found in Table 4.
>
> [1] Lym, Sangkug, et al. "PruneTrain: fast neural network training by dynamic sparse model reconfiguration." In Proc. Int’l Conf. for HPC, Networking, Storage & Analysis, 2019.
>
> [2] Tim Dettmers and Luke Zettlemoyer. “Sparse networks from scratch: Faster training without losing performance.” arXiv, 2019.

---

> > ### Comment · AnonReviewer1 · 2020-11-22
> > **Feedback**
> >
> > Thank you for the details. Answers Q1 & Q2 make sense to me.
> >
> > For Q3, only "BEP 1e-1 and BEP 1e-4 outperform PruneTrain by 0.3% and 0.1%, respectively" under the same inference cost is a solid comparison, however, the improvements are marginal, the method is more complex and introduces more hyper-parameters.
> > Comparison with a lower accuracy but higher efficiency is always hard to judge. A training FLOPs vs accuracy may be more valuable.
> >
> > Moreover, the rebuttal shifts its contribution a bit from training time to inference time reduction. This is tricky. The original tone of the paper is about training time reduction as agreed by other reviewers. In Q2, the authors emphasize its training time reduction since the approach cannot outperform previous pruning methods which of course relying on additional training time. This is OK. However, when it's needed to compare methods for training time reduction (Q3 & Q4), the arguments go to inference time reduction. A figure that will make the claims valid is a plot of training FLOPs vs inference FLOPs for all methods under the same accuracy.

---

> > > ### Author Response · Authors · 2020-11-22
> > > **Review response**
> > >
> > > For Q3. We would like to clarify that BEP 1e-1 reduces the delta from -1.87% to -1.6% (i.e., a 14% reduction in $\Delta$), we believe that this should be stated in context as the original error is quite small already. This was also the highest pruning scenario considered by the PruneTrain paper, thus we cannot test under a higher performance degradation, which would allow for a more significant gap to emerge. We would certainly like to perform further experiments, however the code for PruneTrain is not publicly available.
> > >
> > > Could the reviewer clarify what is meant by, "rebuttal shifts its contribution a bit from training time to inference time reduction?" The inference FLOPs in our problem statement is a user specified argument $B_s$ in Algorithm 1, and is not minimized in our approach. Our contribution is to balance the training time vs. performance tradeoff within a user defined inference FLOPs ($B_s$).
> > >
> > > With regards to Q3/Q4. Respectfully, we acknowledge the value of a perspective focused on training time vs. performance without controlling for inference FLOPs (i.e. $B_s$). However, our contribution is with regards to training time improvement within a user specified inference FLOPs, $B_s$. This derives from extending the pruning problem definition (Eq. 2) along the temporal dimension (Eq. 3/4). We think that there are practical use cases for this approach such as training for resource constrained devices (i.e. edge devices/mobile phones/customer workstations). We hope the reviewer will consider our contribution in view of the use cases we have addressed.

---

> > > ### Author Response · Authors · 2020-11-25
> > > **Newest revision**
> > >
> > > Dear reviewer,
> > >
> > > We have beautified references to PruneTrain within our work to further differentiate our work from PruneTrain. See footnote 4, as well as coverage in related work. We have also added inference FLOPs for both PruneTrain and BEP in Table 6 (Appendix E). We find that BEP $1e-1$ consumes $55.2$% of training FLOPs in contrast to $60$% consumed by PruneTrain. BEP $1e-4$ consumes $56.0$% of training FLOPs.
> > >
> > > We also note in the Appendix E that it is unclear how to solve the early pruning problem (Eq 3,4) with PruneTrain as it does not allow for a mechanism to precisely control the trained network size.

---

### Official Review · AnonReviewer4 · 2020-10-26
**A push in the right direction**

**Rating:** 6
**Confidence:** 5

**Review:**

Summary

This paper introduces a method for pruning during the training process in order to filter out unimportant/redundant components of the network continuously to speed up training and perform gradual pruning over the training process. The proposed approach is novel in the sense that the vast amount of prior work on pruning has focused on either (i) pruning on network initialization (e.g., SNIP, etc.) or (ii) pruning after the network has been fully trained (e.g., Magnitude Pruning, among many others). The introduced method uses the Taylor-series based saliency criterion (of Molchanov et al., 2017) and uses a multi-output Gaussian process to predict future saliencies and to determine whether a parameter can be safely removed early on during training.


Rationale for Score

As far as the negatives go, I take issue with the fact that the proposed approach seems to be highly complicated -- requiring a multitude of hyper-parameters/design choices, and tuning functions/ablation studies (e.g., for lambda). The empirical results are also not very compelling -- the proposed approach requires many more training hours than compared approaches that prune on initialization (cf., SNIP or GRASP in Table 3) and attains only a modest (.3% to .5%) improvement in pruning performance as measured by test accuracy.

With that said, I recommend weak acceptance with the hope that this work inspires more research in this area and that the shortcomings will be remedied in subsequent works that build upon the techniques introduced in this paper. I believe that this work has merit in pushing the community in the direction of trying to achieve one of the overarching goals of pruning: an efficient way to simultaneously train and search for an optimized network architecture for a particular application.


Strengths

- The paper is highly relevant to the ML and optimization communities; the premise that, e.g., filters that would have been pruned anyway after training, should not be trained to save computation time (and improve pruning performance) is very intuitive and appealing.

- It is commendable that the authors tackle the very difficult problem of pruning during training and try to model the interdependencies/future uncertainty in a principled way (using MOGP). To my knowledge, there is no other work that attempts to tackle this problem as rigorously this paper does.

- The method is overall motivated by principled insights and there is some analysis to justify parts of the method (Lemma 1)

- The authors perform evaluations on appropriate benchmarks (ResNet50 trained on ImageNet) and achieve superior pruning results (in terms of test accuracy after training/pruning) relative to those of recently-proposed, state-of-the-art approaches (SNIP and GRASP)


Weaknesses

- The proposed algorithm is not parameter-free (unlike SNIP, which is virtually parameter-free), is quite complicated (and I imagine difficult to implement), and there is little justification for certain components of the method, e.g., the dynamic scaling function (and choices of lambda), whether the simplification of m_{t-1} = … = m_{T} is mild enough. It is not clear to me how a practitioner can run the proposed algorithm in a parameter-free way without having to conduct ablation studies of their own first, especially since, as the authors note, “We observed that the penalty parameter was difficult to tune properly, either being too aggressive at pruning, or too passive” as the justification for the dynamic scaling function

- Parts of the paper are too dense and notation-heavy, and this hurts readability and understanding significantly, e.g., Lemma 1, paragraph regarding the introduction of the saliency function on pg. 2.

- The presented experimental results are not very compelling. For example, in Table 3, we see that BEP 1e-4 achieves a ~.4% improvement over SNIP and GRASP, at the cost of ~7-8.4 more hours of training time. This calls into question the effectiveness of the proposed approach -- which is, at the end of the day, meant to *speed up* training + pruning by removing unnecessary components of the network early on.


Clarity

- The paper is reasonably well-written and organized overall. It was clear that the authors compressed some of the mathematical expressions/lemmas (e.g., statement of Lemma 1), which is somewhat understandable given the page limit, but this hurt readability and understandability.

---

> ### Author Response · Authors · 2020-11-20
> **Review responses**
>
> We thank you for appreciating our contributions and providing valuable feedback. For your concerns about the dense notation and the proposed algorithm being complicated, we have addressed them under the common concerns. We would like to address your remaining comments below:
>
> Q1. Whether the simplification of $m_{t-1} = … = m_{T}$ is mild enough.
>
> We do not think that the simplification $m_{t-1}=, ..., = m_{T}$ is mild. Indeed, this is a coarse approximation that we have to make for resolving the optimization problem in a reasonable time. It is a highly non-trivial problem (at least for us) to look for a mild simplification while remaining competitive in wall time, which we will consider in our future work. Also, it is hard to compare the results after this simplification with the optimal solution since the original optimization problem (i.e., equations 3-4) is too difficult to solve.
>
> Q2. Strength of experimental results.
>
> In our new draft, we have added a new experiment on ResNet-50, pruning a very large percentage of convolutional filters. This may be found in Table 4, rightmost sub-Table. At this setting, BEP 1e-1 (Top-1, 53.7%) outperforms SNIP (50.9%) and GraSP (52.2%). This is much more in line with our results on CIFAR-10/CIFAR-100. Pruning a significant portion of the network is an important use-case as deep learning models continue to grow larger and thus require considerable pruning to allow training on commodity hardware.

---

### Official Review · AnonReviewer2 · 2020-10-30
**Review - Bayesian early pruning**

**Rating:** 6
**Confidence:** 3

**Review:**

This paper introduces a new method to accelerate training by saliency-based pruning. The method predicts future saliency for neurons based on observed saliency with a multi-output Gaussian process (MOGP), then greedily prunes neurons with least saliency at fixed intervals during training. The authors provide extensive mathematical analysis to show that the algorithm produces pruning mask solutions that are close to the optimum of the formulated optimization (the reviewer is unable to verify). The experimental results showed improvements in task accuracies of trained models but with longer training times.

The reviewer believes that the proposed method is novel, as it considers historical statistics during training to provide a more accurate saliency prediction. While this is interesting, the reviewer is concerned with the practicality. The paper can be improved with answers to the following questions:
* The mathematical analysis showed that the algorithm produces pruning mask solutions that are close to the optimum. Is it possible to quantify? The reviewer worries that under a series of approximations and heuristic-based modeling below, the solution no longer aligns with the goal of pruning optimality, and thus makes the mathematical proofs irrelevant:
	* 3.1 problem statement as the pruning objective,
	* 3.2 saliency as MOGP with exponential kernel,
	* 3.3 subsequent simplifications,
	* 4 variational approximation of MOGP.
* What is the overhead introduced by the maximum likelihood estimation of the MOGP, and Bayesian early pruning (BEP, Algorithm 1)? Does the benefit of having a pruned model always outweigh the costs of pruning?
* How accurate is the saliency prediction? Can this be quantified and illustrated somehow, e.g. with the predicted values on Figure 2 in the appendix? If this is not accurate that one may expect the components in the pruning procedure can be replaced with simpler variants without a detrimental impact. For instance, instead of using MOGP for prediction, one may consider static saliency. (Or is this identical to traditional approaches used by SNIP and GRASP?)

Additionally, experiments should not only have averaged results but also provide standard deviations.

The paper is in general well-written, the reviewer has minor complaints:
* In Algorithm 1, `\mu_{T \mid 1:t}` is not assigned value anywhere.

---

> ### Author Response · Authors · 2020-11-20
> **Review responses**
>
> We thank you for providing valuable feedback, which we will take into account when revising our paper. We would like to address your questions below:
>
> Q1. The reviewer worries that under a series of approximations …, and thus makes the mathematical proofs irrelevant:
>
> Firstly, we wish to emphasize that our goal of providing the proofs is to justify our design decisions such that the performance of the approximations can be guaranteed theoretically. About the empirical quantifications, we have added more experimental results in our new draft to show the predictive performance of MOGP (Fig. 1). For other simplifications in Section 3.3, it is highly non-trivial (at least to us) to quantify how close the approximated pruning solution is to the optimal solution empirically since the original optimization problem (i.e., equations 3-4) is too difficult to solve. Now, we can only verify their performance indirectly via the accuracy of the pruned networks. However, we think it is interesting to explore a principled way of quantifying the performance of these simplifications, which we will consider in the future work.
>
> Q2. What is the overhead introduced by the maximum likelihood estimation of the MOGP, and Bayesian early pruning (BEP, Algorithm 1)? Does the benefit of having a pruned model always outweigh the costs of pruning?
>
> In our initial submission, we have presented end-to-end wall time (Table 3) including the time to train the MOGP models. The results show that our proposed BEP (30+ hrs) takes much less time than the ResNet training without pruning (55 hrs). To show the overhead introduced by pruning clearly, we have explicitly delineated the training time into time taken by “training the network” and “time taken by pruning” (e.g., MOGP and pruning steps in BEP) in our new draft. These results can be found in Table 4.
>
> In some cases, we think it is possible for pruning to increase the overall end-to-end training time due to modeling/pruning overhead. Fortunately, the overhead can be reduced by setting a larger value of T_step in our proposed BEP.
>
> Q3. How accurate is the saliency prediction? ... For instance, instead of using MOGP for prediction, one may consider static saliency.
>
> To show the accuracy of the saliency prediction, we added new figures (Fig. 1) in our new draft, which visualize the GP/MOGP predictive mean of the saliency together with the ground-truth saliency values. We can observe that the predictive mean values of MOGP are quite close to the true saliency values. The predictive accuracy of MOGP is much better than that achieved using GP. Also, MOGP is able to capture the long-term trend of saliency curves with significantly less data than GP.
>
>
> Q4. Additionally, experiments should not only have averaged results but also provide standard deviations.
>
> We have provided performance with the standard error in all the results in our new draft. They were previously excluded due to space constraints.
>
> Q5. In Algorithm 1, $\mu_{T \mid 1:t}$ is not assigned value anywhere.
>
> $\mu_{T \mid 1:t}$ is the predictive mean vector computed using MOGP. We have revised Algorithm 1 in our new draft to clarify this and we apologize for the oversight.

---

### Official Review · AnonReviewer3 · 2020-11-02
**Well organized, interesting contribution, experiments could be improved**

**Rating:** 7
**Confidence:** 2

**Review:**

This paper presents a training-time pruning method for deep neural network. The main idea is to include network elements only if they could improve the predictive mean of the saliency, where the saliency measures the efficiency of the network elements in terms of minimizing the loss function.
The paper presents a clear objective of the early pruning, which is to preserve the sub-network that can maximize the saliency function. This optimization problem is NP-hard, and even approximation is very expensive. The authors proves that one can simplify the approximation by ranking the network element by predictive mean of the saliency function.

The authors evaluated their method empirically and showed that it is superior than the GP modeling, and provides a trade-off between accuracy and the training time.

+ This paper is well organized. The theoretical analysis is well written and provides a good review to readers who lack relevant background.
+ The idea is interesting and Lemma 1 should be of value to researchers working on the similar problem.
+ Reading from the review section, the problem of training-time pruning is not well studied yet, whereas this paper could be seen as an important contribution.
- The experiment section is a bit hard to follow. Among other problem, I find it hard to understand the intuition behind the dynamic penalty scaling.
- It is not clear to me why the dynamic sparse reparameterization methods are not listed as baseline in evaluations.

Overall I think this is a good paper. The authors could improve it by addressing the two problems in the evaluation section.

---

> ### Author Response · Authors · 2020-11-20
> **Review responses**
>
> We thank you for providing valuable suggestions and feedback, which we will consider seriously in revising our paper. We have revised the Experiments section so that it is easier to follow. In particular, we would like to address two of your comments below:
>
> Q1. I find it hard to understand the intuition behind the dynamic penalty scaling.
>
> In our new draft (Section 4.2), we have included the intuition behind the dynamic penalty scaling as follows: The dynamic penalty scaling is used to increase the penalty if the anticipated compute required to complete training (i.e.,$(T-t)\lvert\lvert m_t \rvert\rvert_{0}$) begins to exceed the remaining amount of compute budget (i.e., $B_{t, c}$). In such a case, we need to focus more on satisfying the budget constraint (i.e., the second term of equation 6) during the optimization, which is achieved by an increased penalty.
>
> Q2. It is not clear to me why the dynamic sparse reparameterization (DSR) methods are not listed as baseline in evaluations.
>
> We did compare against DSR with respect to small-scale CNN on CIFAR-10/CIFAR-100. The results showed that our proposed BEP better preserves the performance at equivalent network size. We agree that it would be a valuable addition to compare against DSR for more complex networks such as ResNet. However, different from the small-scale experiment which is used to verify only the accuracy of the pruned model, wall training time is an additional important criterion for measuring the performance of the pruning algorithm for a large-scale network. Due to the heterogeneous implementation (PyTorch vs. TensorFlow) of DSR and other tested algorithms, we are not able to make a fair and accurate training time comparison, and thus removed the DSR from the baselines in the ResNet experiments. We are now trying to resolve this implementation issue and will include the DSR results for ResNet in the revised version of this paper.

---

### Author Response · Authors · 2020-11-20
**Common responses**

We humbly and graciously thank the reviewers for understanding our contributions and providing valuable feedback which we will consider seriously in revising our paper. Aside from our individual responses to reviewers, we like to address the following common concerns:

1. Simplifying Notations

In our new draft, we have made the following changes to improve the notation: (a) In Section 2, we have simplified the description of the saliency function and moved its exact definition to the appendix. This does not affect the readability in terms of understanding our proposed BEP algorithm since it is agnostic to the saliency function, as has been mentioned in the Introduction section; (b) We have removed some details regarding MOGP in Section 3.2 and kept only information that is closely related to our pruning formulation; and (c) We have eased the notational clutter in Lemma 1 and made it more readable. To help readers with understanding Lemma 1, we have clarified its implications in the paragraphs after it. We hope these modifications can help the readers to focus on our main novel idea of Early Pruning.

2. On hyperparameters

We would like to clarify that our approach does not have many hyperparameters to be set/tuned. Among the required parameters in Algorithm 1, only two hyperparameters (i.e., lambda and $T_{step}$) need to be set manually; all the others are either user-specified requirements (e.g., budget $B_s$) or parameters with fixed values (e.g., $B_{1,c}$). For these two hyperparameters, we have provided the dynamic penalty scaling strategy for setting lambda automatically and 10-20 works well for $T_{step}$ in practice.

Next, we would like to highlight the practical necessity of dynamic penalty scaling in mitigating the difficulty of deciding an appropriate lambda: Due to the recursive problem definition, an approach to solve Early Pruning must solve several optimization problems ($[\hat{\rho_t}]_{t=1,...,T}$). This, unfortunately, would require the usage of several penalties, one per optimization problem. This abundance of hyperparameters is difficult to tune properly and using a single hyperparameter does not work well in practice. To resolve this, we have chosen to use a feedback loop (dynamic penalty scaling) to determine the penalties dynamically. This feedback loop drives the pruning decisions by modulating the penalty $\lambda_t$ (i.e., $\lambda_{dyn}$ in our initial submission) at iteration t using the feedback on lambda’s efficacy at achieving the desired user-specified sparsity budget $B_s$. Intuitively, the penalty is increased (i.e., the importance of the constraint is increased) if the anticipated compute required to complete training begins to exceed the remaining amount of compute budget. We have shown this derivation more explicitly in Appendix D and cited the relevant literature (PID controllers) which inspired this approach.

In addition, there are secondary hyperparameters (i.e., no. of latent function, variational inducing points) due to our choice of MOGP for saliency modeling as it captures the co-evolution and co-adaptation effects of neural network training. We distinguish BEP hyperparameters from MOGP hyperparameters because in our proposed approach, MOGP may be replaced by any surrogate model which provides a belief over saliency.

We also think that our hyperparameters are robust as similar settings (i.e., derived from our small-scale CIFAR-10/CIFAR-100 experiments) were used in our larger-scale experiments. Thus, we believe that it is possible to use our proposed Algorithm 1 with our recommended settings of hyperparameters and achieve good results.

We agree that it would be interesting to see how $T_{step}$ and the various MOGP parameters would affect the final pruned result, though we have performed validation which indicated relatively good hyperparameter settings for the number of latent functions ranging between 4 and 18, as well as a performance improvement of MOGP modeling over GP modeling (Table 1). In our new draft, we have included a preliminary ablation study which varies $T_{step}$ as well as the variational inducing points of our MOGP model (Table 3). We find that these two parameters are fairly robust to changes. We will attempt to expand this study further within the time of the discussion period.

---

### Author Response · Authors · 2020-11-25
**Final revision**

Dear reviewers,

As the discussion period closes, we have one last revision for you. We have completed our ablation study, varying the hyperparameters of our approach. This can be found in Table 3. We find that in general, all hyperparameters are robust to changes with mild degradation observed only in the extremal settings.

We have also added a slight beautifying change to further differentiate our use case/problem definition from PruneTrain [1] with footnote 4 in the problem statement.

We hope that you will keep the above in mind during internal discussion. Once again, we extend our most humble and gracious thanks for your valuable review and feedback.

Sincerely,

The authors of “Balancing training time vs. performance with Bayesian Early Pruning.”

[1] Lym, Sangkug, et al. "PruneTrain: fast neural network training by dynamic sparse model reconfiguration." In Proc. Int’l Conf. for HPC, Networking, Storage & Analysis, 2019.

---

### Decision · Program_Chairs · 2021-01-07
**Final Decision**

**Decision:**

Reject

**Comment:**

This paper considers the problem of pruning deep neural networks (DNNs) during training. The key idea is to include DNN elements only if they improve the predictive mean of the saliency (efficiency of the DNN elements in terms of minimizing the loss function). The objective of early pruning is to preserve the sub-network that can maximize saliency. This optimization problem is NP-hard, and even approximation is very expensive. The paper proves that one can simplify the approximation by ranking the network element by predictive mean of the saliency function.

The proposed approach is novel as most of the prior work on pruning has focused on either (i) pruning on network initialization  or (ii) pruning after the network has been fully trained.

Couple of issues with the paper are:
1. Current approach is somewhat complicated with many hyper-parameters
2. Experimental results are not very compelling when compared to pruning on network initialization

Overall, my assessment is that the paper takes a new research direction and has the potential to inspire the community, and followup work may be able to overcome the above two issues in future. However, due to the remaining shortcomings, the paper is not judged ready for publication in its present form. I strongly encourage to resubmit the paper after addressing the above two concerns.